# Deciphering the genetic code of neuronal type connectivity through bilinear modeling

Mu Qiao*

LinkedIn, Mountain View, United States

**Abstract** Understanding how different neuronal types connect and communicate is critical to interpreting brain function and behavior. However, it has remained a formidable challenge to decipher the genetic underpinnings that dictate the specific connections formed between neuronal types. To address this, we propose a novel bilinear modeling approach that leverages the architecture similar to that of recommendation systems. Our model transforms the gene expressions of presynaptic and postsynaptic neuronal types, obtained from single-cell transcriptomics, into a covariance matrix. The objective is to construct this covariance matrix that closely mirrors a connectivity matrix, derived from connectomic data, reflecting the known anatomical connections between these neuronal types. When tested on a dataset of *Caenorhabditis elegans*, our model achieved a performance comparable to, if slightly better than, the previously proposed spatial connectome model (SCM) in reconstructing electrical synaptic connectivity based on gene expressions. Through a comparative analysis, our model not only captured all genetic interactions identified by the SCM but also inferred additional ones. Applied to a mouse retinal neuronal dataset, the bilinear model successfully recapitulated recognized connectivity motifs between bipolar cells and retinal ganglion cells, and provided interpretable insights into genetic interactions shaping the connectivity. Specifically, it identified unique genetic signatures associated with different connectivity motifs, including genes important to cell-cell adhesion and synapse formation, highlighting their role in orchestrating specific synaptic connections between these neurons. Our work establishes an innovative computational strategy for decoding the genetic programming of neuronal type connectivity. It not only sets a new benchmark for single-cell transcriptomic analysis of synaptic connections but also paves the way for mechanistic studies of neural circuit assembly and genetic manipulation of circuit wiring.

*For correspondence:
muqiao0626@gmail.com

Competing interest: The author declares that no competing interests exist.

## eLife assessment

This is an **important** computational study that applies the machine learning method of bilinear modeling to the problem of relating gene expression to connectivity. Specifically, the author attempts to use transcriptomic data from mouse retinal neurons to predict their known connectivity with promising results. On revision, the approach was tested against a second data set from *C. elegans*. A limited number of genes studied in this second dataset may have resulted in performance that matched but did not exceed prior models. However, taken together, the results were felt to provide **solid** evidence for the value of the approach.

## Introduction

One of the fundamental objectives in neuroscience is understanding how diverse neuronal cell types establish connections to form functional circuits. This understanding serves as a cornerstone for decoding how the nervous system processes information and coordinates responses to stimuli (*Seung,*

*2012*). Despite this, the genetic mechanisms determining the specific connections between distinct neuronal types, especially within complex brain structures, remains elusive (*Polleux and Snider, 2010*; *Sanes and Zipursky, 2020*).

Recent advances in transcriptomics and connectomics provide opportunities to probe this. Single-cell transcriptomics enables high-resolution profiling of gene expressions across neuronal types (*Zeng and Sanes, 2017*; *Stegle et al., 2015*), while connectomic data offers detailed maps quantifying connections between neuronal cell types (*Denk and Horstmann, 2004*; *Helmstaedter et al., 2013*; *Tapia et al., 2012*). However, the challenge of linking gene expressions derived from single-cell transcriptomics to neuronal type connectivity evident from connectomic data to uncover the genetic underpinnings has yet to be fully addressed.

Drawing inspiration from the field of machine learning, particularly recommendation systems, we introduce a bilinear model to bridge this gap. This model, in the context of recommendation systems, has been successful in capturing intricate user-item interactions (*Koren et al., 2009*). By treating the gene expressions of pre- and post-synaptic neurons and their connectivity akin to users, items, and their ratings, we adapt the architecture of recommendation systems to the neurobiological domain. We hypothesize that a similar model could capture the complex relationships between genetic patterns of presynaptic and postsynaptic neurons and their connectivity.

This bilinear modeling approach was first applied to a *Caenorhabditis elegans* (*C. elegans*) neuronal dataset, where it not only matched but slightly outperformed the spatial connectome model (SCM) in reconstructing the connectivity of electrical synapses or gap junctions from innexin gene expressions. Notably, it revealed additional genetic interactions beyond those uncovered by the SCM. When extended to mouse retinal neurons, we demonstrate that it could effectively reconstruct synaptic connectivity between bipolar cells (BCs) and retinal ganglion cells (RGCs) from their gene expressions. The model not only unveils connectivity motifs between BCs and RGCs but also provides biologically meaningful insights into candidate genes and the genetic interactions that orchestrate this connectivity. Furthermore, our model predicts potential BC partners for RGC transcriptomic types, with these predictions aligned substantially with functional descriptions of these cell types from previous studies. Collectively, this work significantly contributes to the ongoing exploration of the genetic code underlying neuronal connectivity and suggests a potential paradigm shift in the analysis of single-cell transcriptomic data in neuroscience.

## Background

### Synaptic specificity

The intricate neural networks that form the basis of our nervous system are a product of specific synaptic connections between different types of neurons. This specificity is not a mere coincidence but a meticulously orchestrated process that underpins the functionality of the entire network (*Sanes and Zipursky, 2020*; *Martin et al., 2020*). Each neuron can form thousands of connections, or synapses, with other neurons, and the specificity of these connections determines the neuron's function and, by extension, the network's function as a whole.

Synaptic specificity encompasses both chemical synapses, which rely on neurotransmitter-mediated communication between pre- and post-synaptic neurons (*Sanes and Zipursky, 2020*), and electrical synapses, where direct transmission of ions or small molecules occurs via gap junctions (*Martin et al., 2020*). A classic example of chemical synaptic specificity is observed in the retina, where different types of BCs form specific synaptic connections with various types of RGCs (*Helmstaedter et al., 2013*; *Euler et al., 2014*; *Sanes and Masland, 2015*). These connections create parallel pathways that transform visual signals from photoreceptors to RGCs, which subsequently transmit the information to the brain (*Gollisch and Meister, 2010*; *Azeredo da Silveira and Roska, 2011*). Meanwhile, specific gap junction connections, composed of connexins in vertebrates and innexins in invertebrates, has been observed between *C. elegans* neurons (*Kumar and Gilula, 1996*; *Phelan et al., 1998*; *Rabinowitch and Schafer, 2015*; *Marcus et al., 2014*; *Südhof, 2017*). They function broadly in neural circuits of sensory processing and behavioral output (*Martin et al., 2020*; *Hall, 2017*).

The genetic principles guiding the formation of these specific connections, particularly in complex brain structures, remains elusive. The brain's complexity, with its billions of neurons and trillions of synapses, poses significant challenges in identifying the specific genes and genetic mechanisms that guide the formation of these connections. Despite advances in genetic and neurobiological research,

such as understanding the roles of certain recognition molecules and adhesion molecules in synaptic specificity, the genetic foundation of connectivity between neuronal types is still largely unknown (*Sanes and Zipursky, 2020*; *de Wit and Ghosh, 2016*; *Martin et al., 2020*).

Emerging tools and technologies offer unprecedented opportunities to unravel these mysteries. Among these, transcriptome and connectome are particularly promising (*Sanes and Zipursky, 2020*; *Fornito et al., 2019*). Transcriptome, the complete set of RNA transcripts produced by the genome, can provide valuable insights into the genes that are active in different types of neurons and at different stages of neuronal development. This can help identify candidate genes that may play a role in guiding neuronal connectivity. Connectome, on the other hand, provides a detailed map of the connections between neurons. By combining information from transcriptome and connectome, it is possible to link specific genes to specific connections, thereby shedding light on the genetic basis of synaptic connectivity.

## Previous approaches

Prior research has reported several methodologies to unravel the genetic underpinnings of neuronal connectivity. For instance, Kaufman et al. showed a correlation between gene expression of *C. elegans* neurons and their connectivity (*Kaufman et al., 2006*), and Varadan et al. developed an entropy minimization approach for understanding the molecular logic of synaptic connectivity in *C. elegans* (*Varadan et al., 2006*). These models, however, did not fully account for spatial constraints for synaptic formation.

In response, subsequent studies proposed methodologies that integrate gene expressions with neuronal connectivity, taking into consideration physical contacts between neurons (*Kovács et al., 2020*; *Barabási and Barabási, 2020*; *Taylor et al., 2021*). Specifically, the SCM in Kovács et al. correlates the gene expression of neurons with their connectivity via a rule matrix. This model aims to minimize the discrepancy between predicted connectivity based on gene expression, and observed connectivity. By restricting the analysis to neuron pairs that are in physical contact, the SCM transforms the original problem into a regression between the Kronecker product of the gene expression matrix and an edge list that captures neuronal connectivity (*Kovács et al., 2020*).

Additionally, Taylor et al. introduced the network differential gene expression analysis (nDGE), a statistical method that expands upon traditional differential gene expression analysis by examining the co-expression of gene pairs between neuron pairs, comparing synaptic versus non-synaptic neuronal groups through t-tests. It incorporates physical contacts between neurons through the generation of 'pseudoconnectomes' for null distribution estimation. Unlike multivariate methods such as the SCM, nDGE operates as a mass-univariate method, focusing on single gene pairs' contributions to synaptic formation without considering the complex interactions among multiple co-expressed genes. This makes nDGE's findings inherently conservative, ensuring strict control over type 1 errors but potentially underestimating the multifaceted nature of synaptic connectivity (*Taylor et al., 2021*).

While the SCM and nDGE models have focused on the connectivity of individual neurons and were tested using *C. elegans* datasets, their generalization to neuronal cell types has not been explored. As we move from the invertebrate nervous systems to the neural architectures of vertebrates, such as those in mice or macaques, we need methodologies capable of unraveling the genetic basis of neuronal type connectivity (*Zeng and Sanes, 2017*; *Zeng, 2022*).

## Collaborative filtering

Our strategy draws inspiration from the concept of collaborative filtering using bilinear models, a technique fundamental to recommendation systems (*Ricci et al., 2011*; *Su and Khoshgoftaar, 2009*). These systems predict a user's preference for an item (e.g. a movie or product) based on user-item interaction data.

Bilinear models capture the interaction between users and items via low-dimensional latent features (*Koren et al., 2009*; *Rendle et al., 2012*). Mathematically, for user $i$ and item $j$, we denote their original features as $x_i \in \mathbf{R}^{1 \times p}$ and $y_j \in \mathbf{R}^{1 \times q}$, respectively. These features are then projected into a shared latent space with dimension $d$ via transformations $x_i A$ (where $A \in \mathbf{R}^{p \times d}$) and $y_j B$ (where $B \in \mathbf{R}^{q \times d}$). The predicted rating of the user for the item is then formulated as:

$$r_{ij} = (x_i A)(y_j B)^T \tag{1}$$

In the context of collaborative filtering, the goal is to optimize the transformation matrices $A$ and $B$ to align the predicted rating $r_{ij}$ with the ground-truth $z_{ij}$. This is expressed as the following optimization problem:

$$\min_{A,B} \sum_{ij} (z_{ij} - (x_i A)(y_j B)^T)^2 \tag{2}$$

Or in the matrix form:

$$\min_{A,B} \|Z - (XA)(YB)^T\|_F^2 \tag{3}$$

Here, the objective is to minimize the Frobenius norm of the residual matrix $Z - (XA)(YB)^T$.

In our study, we interpret neuronal connectivity through the lens of recommendation systems, viewing presynaptic neurons as 'users', postsynaptic neurons as 'items', and the synapses formed between them as 'ratings'. Our chosen bilinear model extracts latent features of pre- and post-synaptic neurons from their respective gene expressions. One key advantage of the bilinear model is its capacity to assign different weights to the gene expressions of pre- and post-synaptic neurons, enabling the model to capture not just homogeneous but also complex, heterogeneous interactions fundamental to understanding neuronal connectivity. Prior studies have highlighted such heterogeneous interactions, noting the formation of connections between pre- and post-synaptic neurons expressing different cadherins, indicative of a heterogeneous adhesion process (*Duan et al., 2014*; *Duan et al., 2018*).

## Results

### Bilinear model for neuronal type connectivity

We discuss the bilinear model for neuronal type connectivity in the following two scenarios: the first in which gene expression and connectivity of each cell are known simultaneously and the second where connectivity and gene expressions of neuronal types are from different sources. The bilinear models for these two situations are illustrated in *Figure 1*.

#### Gene expression and connectivity of each cell are known simultaneously

We begin with an ideal scenario where both the gene expression profiles and connectivity of individual cells are known concurrently. In this setting, we have $a$ presynaptic neuronal types and $b$ postsynaptic neuronal types, indexed by $i$ and $j$, respectively. Each type contains a number of neurons, signified as $n_i$ for presynaptic and $n_j$ for postsynaptic types. The gene expression vector for the $k^{th}$ cell in the presynaptic type $i$ is designated as $x_{(ik)}$, where $k \in 1, 2, ..., n_i$, while for the $l^{th}$ cell in postsynaptic type $j$, it is $y_{(jl)}$ with $l \in 1, 2, ..., n_j$. We depict the connectivity metric between a presynaptic neuron and a postsynaptic neuron as $z_{(ik)(jl)}$.

Drawing from the principles of collaborative filtering, we develop the following optimization objective:

$$\min_{A,B} \sum_{i=1}^{a} \sum_{j=1}^{b} \left( \frac{1}{n_i n_j} \sum_{k=1}^{n_i} \sum_{l=1}^{n_j} (z_{(ik)(jl)} - (x_{(ik)}A)(y_{(jl)}B)^T)^2 \right) \tag{4}$$

Here, $A$ and $B$ denote the transformation matrices we aim to learn. This formula can also be expressed in its matrix form as:

$$\min_{A,B} \|W \odot (Z - (XA)(YB)^T)\|_F^2 \tag{5}$$

In this equation, $W$ symbolizes a weight matrix where each element $w_{(ik)(jl)} = \frac{1}{\sqrt{n_i n_j}}$. As our study focuses on the genetic code of pre- and post-synaptic neuronal types rather than individual neurons, this weight matrix ensures that the model does not disproportionately favor neuronal types with a greater number of neurons over rarer types. Note that this formulation can be generalized to individual cell level analysis by treating each cell as a type and setting $n_i = n_j = 1$, thus allowing exploration of genetic underpinnings of connectivity at the single-cell resolution.

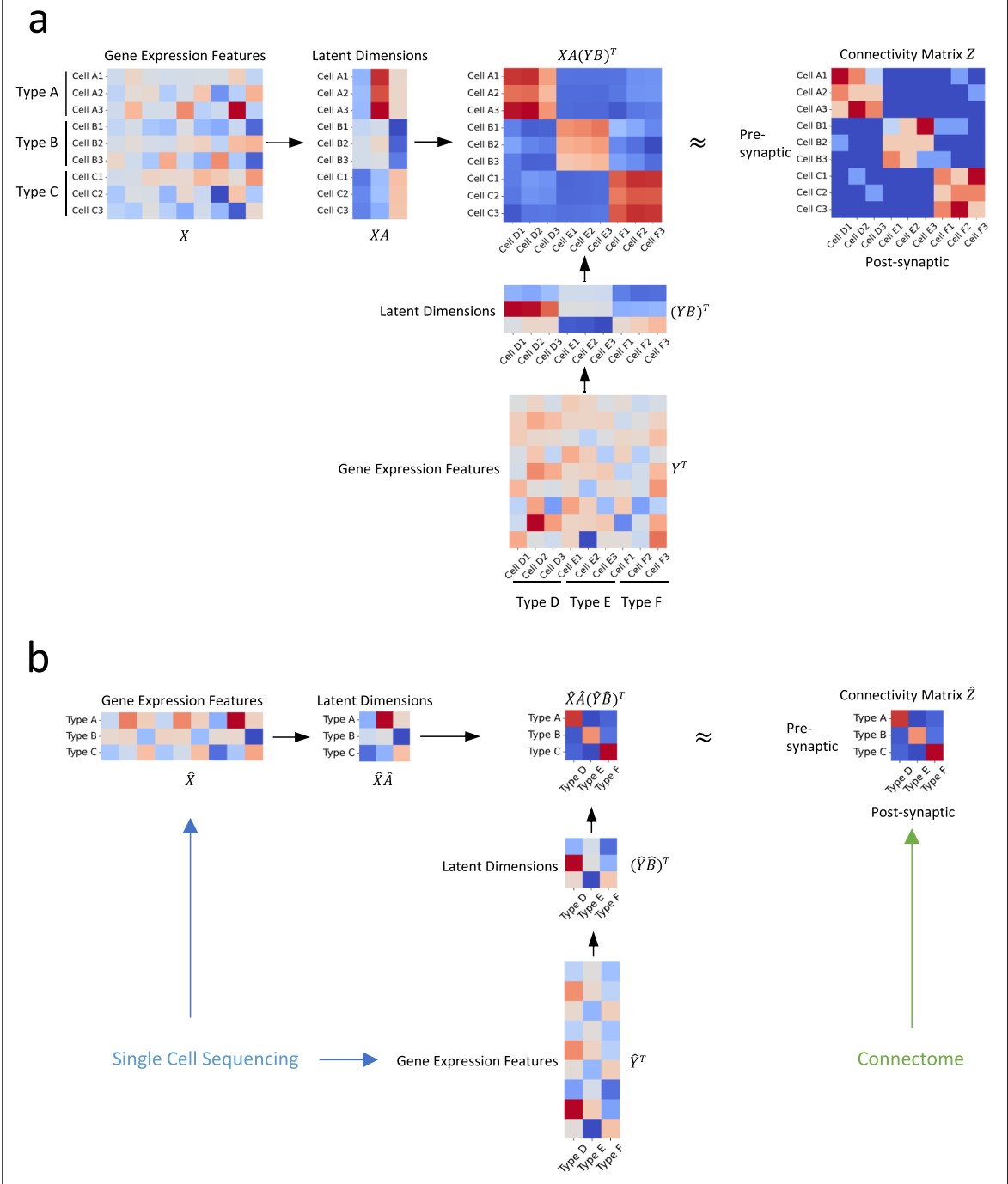

**Figure 1.** Illustration of our approach. (**a**) In an ideal scenario where gene expression profiles and connectivity data of individual cells are available simultaneously, we establish the relationship between connectivity and gene expression profiles via two transformation matrices $A$ and $B$ (**b**) In practical situations where the gene expression profiles are derived from distinct sources, such as single-cell transcriptomic and connectomic data, we propose that the connectivity of individual cells and their latent gene expression features can be approximated by the averages of their corresponding cell types, and establish their relationship through transformation matrices $\hat{A}$ and $\hat{B}$.

In the context of high dimensionality of gene expressions, the bilinear model may face a common issue in machine learning called multicollinearity, a condition where one or more predictor variables are highly correlated. To mitigate this, we can perform principal component analysis (PCA) on the gene expression vectors, transforming them into a new coordinate system and removing components with negligible eigenvalues to reduce redundant information. Alternatively, we can apply regularization

techniques, such as L2 regularization (Ridge) or L1 regularization (Lasso) to effectively manage the multicollinearity. These regularization methods work by imposing a penalty on the size of the linear coefficients in the model, thereby shrinking the coefficients and stabilizing their estimates.

Incorporating L2 regularization, we minimize the following loss function with regularization hyperparameters $\lambda_A$ and $\lambda_B$:

$$L(A, B) = \|W \odot (Z - (XA)(YB)^T)\|_F^2 + \frac{\lambda_A}{2}\|A\|_F^2 + \frac{\lambda_B}{2}\|B\|_F^2 \tag{6}$$

To optimize this function, we propose an alternative gradient descent algorithm. This algorithm alternates between updating the transformation matrices $A$ and $B$, using the gradient descent optimization method.

The algorithm begins by initializing transformation matrices $A$ and $B$ using random values drawn from a standard normal distribution. The central aspect of the algorithm is an iterative loop that alternates the updates of $A$ and $B$. During each iteration, the algorithm first computes the predicted connectivity metric $Z$ using the current estimates of $A$ and $B$. Subsequently, the gradient of the loss function with respect to the transformation matrices is calculated, and the matrices are updated by moving in the negative gradient's direction. This iterative process is repeated until the transformation matrices $A$ and $B$ converge to a steady solution. Upon completion, the algorithm yields the optimized transformation matrices.

This gradient descent-based algorithm provides a computationally efficient solution to the bilinear mapping problem between gene expression profiles and connectivity metrics. As a result, it produces associations between gene expression profiles of cell types and their connectivity.

---

**Algorithm 1.** **Alternative Gradient Descent (AGD) for 'Gene expression and connectivity of each cell are known simultaneously'**

---

1: **Procedure** AGD $(Z, X, Y, d, r, \lambda_A, \lambda_B)$ $\triangleright d$: latent space dimension; $r$: learning rate
2: $q \leftarrow$ second dimension of $X$
3: $p \leftarrow$ second dimension of $Y$
4: Initialize $A$ with random values of size $(q, d)$
5: Initialize $B$ with random values of size $(p, d)$
6: **while** not converged **do**
7: $\hat{Z} \leftarrow XA(YB)^T$ $\triangleright \hat{Z}$: prediction of $\bar{Z}$
8: Compute $A_{\text{grad}} \leftarrow 2X^T(W \odot (\hat{Z} - Z))YB + \lambda_A A$
9: Update $A \leftarrow A - r * A_{\text{grad}}$
10: Compute $B_{\text{grad}} \leftarrow 2Y^T(W \odot (\hat{Z} - Z))^T XA + \lambda_B B$
11: Update $B \leftarrow B - r * B_{\text{grad}}$
12: **end while**
13: **return** $A, B$
14: **end procedure**

---

## Connectivity and gene expressions of neuronal types are from different sources

In real scenarios, gene expression profiles and connectivity information are often derived from separate sources, such as single-cell sequencing (*Shekhar et al., 2016*; *Tran et al., 2019*) and connectome data (*Helmstaedter et al., 2013*; *Bae et al., 2018*; *Greene et al., 2016*). Bridging these datasets requires classifying neurons into cell types based on their gene expression profiles and morphological characteristics. These cell types from different sources are subsequently aligned according to established biological knowledge (e.g. specific gene markers are known to be expressed in certain morphologically defined cell types *Goetz et al., 2022*).

The primary challenge in this scenario is that, while we can align cell types (denoted by indices $i$ and $j$ in *Equation 4*), we are unable to associate individual cells (represented by indices $k$ and $l$ in *Equation 4*). To tackle this issue, we adopt a simplifying assumption that the connectivity and latent gene expression features of individual cells can be approximated by the averages of their corresponding cell types. This premise hinges on the notion that the connectivity metrics and latent gene expression features of individual cells are close enough to the mean value of their corresponding cell types.

As a result, our optimization objective in *Equation 4* becomes:

$$\min_{A,B} \sum_{i=1}^{a} \sum_{j=1}^{b} (z_{(i.)(j.)} - (x_{(i.)}A)(y_{(j.)}B)^T)^2 \tag{7}$$

In this equation, $z_{(i.)(j.)}$ denotes the mean connectivity metric between presynaptic cell type $i$ and postsynaptic cell type $j$. Meanwhile, $x_{(i.)}$ and $y_{(j.)}$ represent the average gene expressions of cell types $i$ and $j$ respectively.

While optimizing the transformation matrices $A$ and $B$, we impose constraints on these matrices to ensure that the variance of latent gene expression features within each neuronal type is minimized. Specifically, we define $\epsilon$ as a small enough value and impose the following constraints on $A$:

$$\|A^T \Sigma_x A\|_F^2 \leq \epsilon \tag{8}$$

where

$$\Sigma_x = \sum_{i=1}^{a} \left( \frac{1}{n_i} \sum_{k=1}^{n_i} (x_{(ik)} - x_{(i.)})^T (x_{(ik)} - x_{(i.)}) \right) \tag{9}$$

and $B$:

$$\|B^T \Sigma_y B\|_F^2 \leq \epsilon \tag{10}$$

where

$$\Sigma_y = \sum_{j=1}^{b} \left( \frac{1}{n_j} \sum_{l=1}^{n_j} (y_{(jl)} - y_{(j.)})^T (y_{(jl)} - y_{(j.)}) \right) \tag{11}$$

These conditions assure that the latent gene expression features of individual cells are proximate enough to the average value within their respective cell types. With these constraints in mind, we formulate the optimization problem as follows:

$$\min_{A,B} \|\bar{Z} - \bar{X}A(\bar{Y}B)^T\|_F^2, \quad s.t. \|A^T \Sigma_x A\|_F^2 \leq \epsilon, \|B^T \Sigma_y B\|_F^2 \leq \epsilon \tag{12}$$

In this equation, $\bar{X} \in \mathbf{R}^{a \times p}$ denotes the average gene expressions of the $a$ presynaptic cell types, wherein each element $\bar{x}_{im}$ is indicative of the average gene expression feature $m$ within cell type $i$. Likewise, $\bar{Y} \in \mathbf{R}^{b \times q}$ represents the average gene expressions of the $b$ postsynaptic cell types, with each element $\bar{y}_{jm}$ signifying the average gene expression feature $m$ in cell type $j$.

In practical application, we approximate $\Sigma_x$ and $\Sigma_y$ with their diagonal estimates $diag(\hat{\sigma}_{x_1}^2, \hat{\sigma}_{x_2}^2, ..., \hat{\sigma}_{x_p}^2)$ and $diag(\hat{\sigma}_{y_1}^2, \hat{\sigma}_{y_2}^2, ..., \hat{\sigma}_{y_q}^2)$(**Butler et al., 2018**; **Stuart et al., 2019**). We then transform the initial optimization problem into the following:

$$\min_{\hat{A},\hat{B}} \|\bar{Z} - \hat{X}\hat{A}(\hat{Y}\hat{B})^T\|_F^2, \quad s.t. \|\hat{A}^T \hat{A}\|_F^2 \leq \epsilon, \|\hat{B}^T \hat{B}\|_F^2 \leq \epsilon \tag{13}$$

where elements in $\hat{X} \in \mathbf{R}^{a \times p}$ are defined as $\hat{x}_{im} = \frac{\bar{x}_{im}}{\hat{\sigma}_{x_m}}$ and elements in $\hat{Y} \in \mathbf{R}^{b \times q}$ are given by $\hat{y}_{im} = \frac{\bar{y}_{im}}{\hat{\sigma}_{y_m}}$. The optimization of this formulation tends to be computationally more tractable.

Here, instead of aligning at the level of individual cells, we focus on the alignment of neuronal types. We achieve this by mapping gene expressions into a latent space via transformation matrices $\hat{A}$ and $\hat{B}$, with the optimization process aiming to minimize the discrepancies between these two sources of information while maintaining consistency of the gene expression features within individual neuronal types.

To solve the optimization problem as outlined in **Equation 13**, we construct the following loss function:

$$L(\hat{A}, \hat{B}) = \|\bar{Z} - \hat{X}\hat{A}(\hat{Y}\hat{B})^T\|_F^2 + \frac{\lambda_A}{2} \|\hat{A}^T \hat{A}\|_F^2 + \frac{\lambda_B}{2} \|\hat{B}^T \hat{B}\|_F^2 \tag{14}$$

where $\lambda_A$ and $\lambda_B$ are hyperparameters whose optimal values are determined through a grid search.

To optimize this loss function, we employ an alternative gradient descent algorithm analogous to that described above, by iteratively updating the transformation matrices $\hat{A}$ and $\hat{B}$.

---

**Algorithm 2. Alternative Gradient Descent (AGD) for 'Connectivity and gene expressions of neuronal types are from different sources'**

---

1: Procedure AGD($\bar{Z}, \hat{X}, \hat{Y}, d, r, \lambda_A, \lambda_B$)  ▷$d$: latent space dimension; $r$: learning rate
2:   $q \leftarrow$ second dimension of $\hat{X}$
3:   $p \leftarrow$ second dimension of $\hat{Y}$
4:   Initialize $\hat{A}$ with random values of size $(q, d)$
5:   Initialize $\hat{B}$ with random values of size $(p, d)$
6:   **while** not converge **do**
7:     $\hat{Z} \leftarrow \hat{X}\hat{A}(\hat{Y}\hat{B})^T$  ▷$\hat{Z}$: prediction of $\bar{Z}$
8:     Compute $\hat{A}_{grad} \leftarrow \hat{X}^T(\hat{Z} - \bar{Z})\hat{Y}\hat{B} + \lambda_A\hat{A}(\hat{A}^T\hat{A})$
9:     Update $\hat{A} \leftarrow \hat{A} - r * \hat{A}_{grad}$
10:    Compute $\hat{B}_{grad} \leftarrow \hat{Y}^T(\hat{Z} - \bar{Z})\hat{X}\hat{A} + \lambda_B\hat{B}(\hat{B}^T\hat{B})$
11:    Update $\hat{B} \leftarrow \hat{B} - r * \hat{B}_{grad}$
12:  **end while**
13:  return $\hat{A}$, $\hat{B}$
14: end procedure

---

## Comparative analysis of bilinear model and SCM of using *C. elegans* neuronal data

### Reconstruction of *C. elegans* gap junction connectivity from innexin expressions

Utilizing the *C. elegans* neuronal dataset, we first tried to reconstruct the gap junction connectivity network based solely on the expression profiles of innexin genes. Using $A$ and $B$ generated by the bilinear model, we processed the innexin expression data to predict gap junction connectivity between neuron pairs as $XA(YB)^T$ (*Figure 2a*). This approach was then compared to the SCM proposed by *Kovács et al., 2020*, which used a rule matrix $O$ to correlate gene expression with observed connectivity in the form of $XOX^T$ (*Figure 2b*).

The effectiveness of both models was evaluated against the observed gap junction connectivity matrix of *C. elegans* neurons (*Figure 2c*). Given the binary nature of the ground truth matrix (where 1 denotes a connection and 0 indicates its absence) and the continuous nature of reconstructed connectivity matrices from both models, we conducted Receiver Operating Characteristic (ROC) analysis. This involves varying a threshold to binarize the continuous predictions, under which the true positive rate is plotted against the false positive rate for each possible cutoff. This process yields the ROC curve, which is a graphical representation of the trade-off between sensitivity and specificity at various thresholds (*Figure 2d*).

Subsequently, we calculated the Area Under the ROC Curve (AUC), providing a singular value summarizing the overall predictive performance of the model across all thresholds. The ROC-AUC metric is particularly informative as it aggregates the model's effectiveness over all possible thresholds, with a score of 1 indicating perfect prediction and 0.5 denoting a performance no better than random chance. From the calculation, the bilinear model achieved a ROC-AUC score of 0.6435, slightly surpassing the SCM's score of 0.6428. While both scores are reasonably close, the slight edge of the bilinear model indicates its nuanced efficiency in mapping gene expressions to connectivity. However, it is noteworthy that both scores, while above 0.5, are substantially distant from the ideal score of 1. This observation suggests that relying exclusively on innexin expression data might be insufficient for fully capturing the detailed gap junction connectivity in *C. elegans*.

### Comparison of rule matrix from SCM and bilinear transformation matrices

In light of the challenge in fully capturing the *C. elegans* gap junction connectivity based on innexin expression data alone, instead of analyzing connectivity motifs between *C. elegans* neurons, our focus pivoted towards exploring and comparing the genetic rules inferred by both the bilinear model and the SCM, which was also the key discussion presented in *Kovács et al., 2020*. As mentioned in '*C. elegans* neuronal dataset' and discussed in the Disscussion, the product of the bilinear transformation

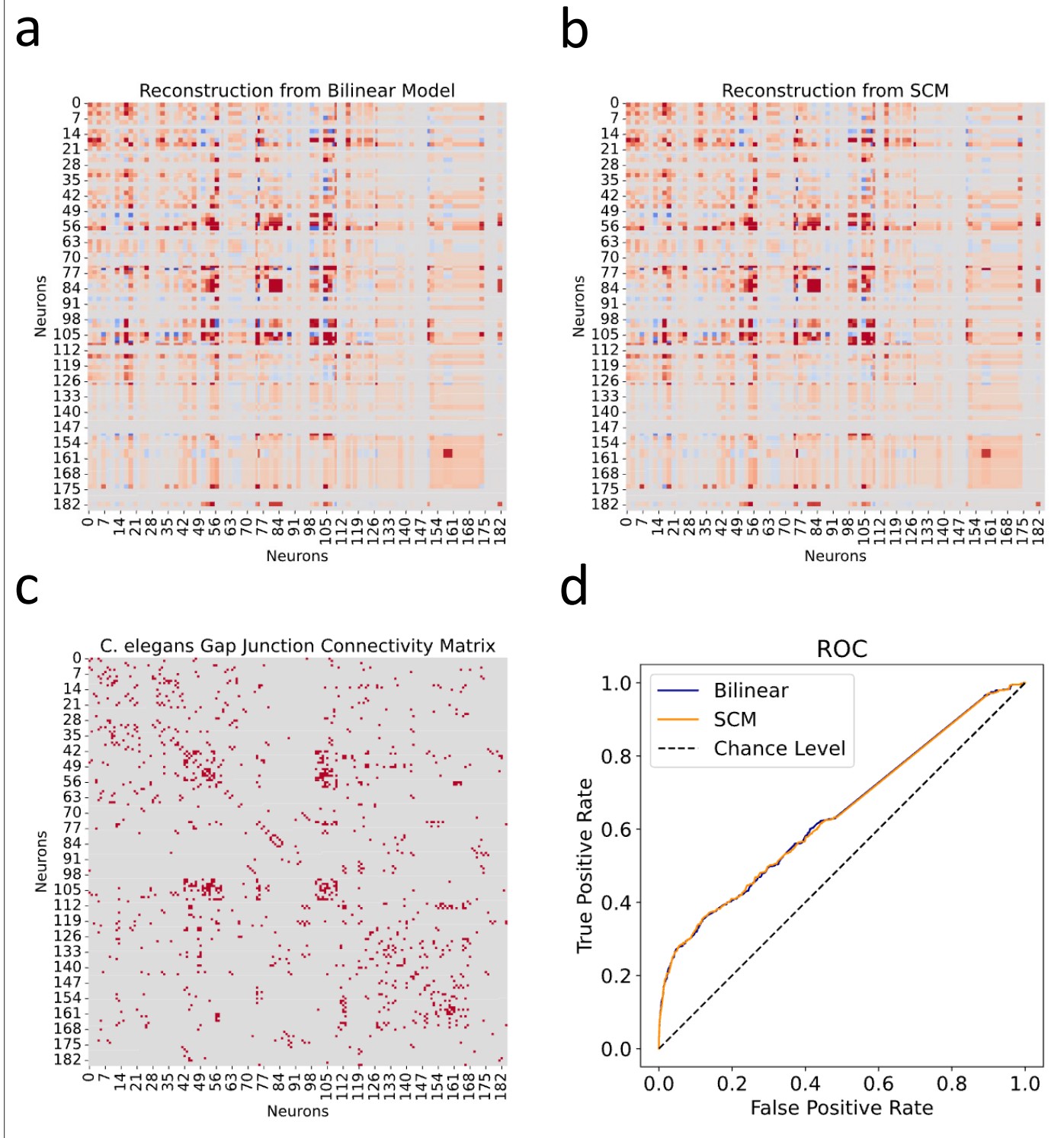

**Figure 2.** Reconstructed gap junction connectivity from innexin expression data. (**a**) Connectivity matrix predicted by the bilinear model. (**b**) Connectivity matrix modeled from Kovács et al.'s SCM. (**c**) Observed gap junction connectivity matrix, serving as ground truth. The color spectrum from red to gray denotes the spectrum from strong connections to weak or no connections. (**d**) ROC curves from both the bilinear model and the SCM. Dashed line indicates the chance level.

The online version of this article includes the following figure supplement(s) for figure 2:

**Figure supplement 1.** Hyperparameter selection through cross-validation for the *C. elegans* neuronal dataset.

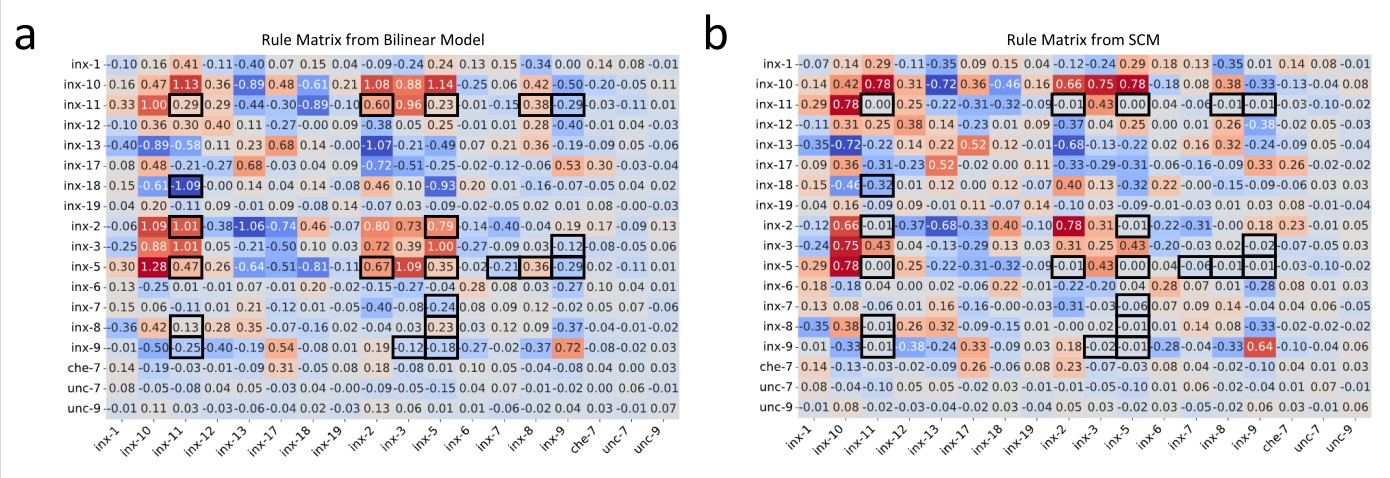

**Figure 3.** Genetic rules from the bilinear model and the SCM. (**a**) The rule matrix $\boldsymbol{AB}^T$ derived from the bilinear model. (**b**) The rule matrix $\boldsymbol{O}$ from the SCM. Black boxes highlight entries with substantial differences.

The online version of this article includes the following figure supplement(s) for figure 3:

**Figure supplement 1.** Detailed discrepancy analysis between the bilinear model and SCM genetic rules.

matrices, $\hat{\boldsymbol{O}} = \boldsymbol{AB}^T$, can be interpreted as a lower-dimensional reconstruction of the rule matrix $\boldsymbol{O}$ used in the SCM. This perspective steered us to a meticulous comparative analysis between the two matrices.

The rule matrix solved from the SCM establishes a baseline for the comparison (*Figure 3b*). Against that, we compared the product of the bilinear transformation matrices (*Figure 3a*). Visualization of the two matrices suggests a high degree of similarity between them, which is quantitatively supported by a Pearson correlation coefficient of 0.90 (p < 0.001), underscoring a strong alignment.

To discern specific genetic interactions uniquely characterized by each model, we applied the DS metric to corresponding matrix entries (*Figure 3—figure supplement 1a*; see 'Methods and supplementary materials' for details). This metric, ranging from 0 (no discrepancy) to 1 (maximum discrepancy), was thresholded at 0.5 to highlight entries with substantial differences. Further, to account for the regularization effect that pushes less important coefficients toward zero, we filtered out entry pairs where both values were less than 0.1 (*Figure 3—figure supplement 1b and c*). The remaining pairs are highlighted in black boxes in both matrices (*Figure 3*).

Comparing the values of highlighted entry pairs, we found that the bilinear model not only captured all genetic interactions identified by the SCM but also inferred additional ones: certain innexins (inx-11, inx-8, inx-5, and inx-2) were implicated in co-expression patterns within connected neurons, while another set (inx-11, inx-9, inx-3, inx-5, inx-7) was associated with an avoidance pattern, suggesting a lack of co-expression in neuron pairs forming gap junctions. These findings provide extra candidates to be tested in future experiments.

### Application of bilinear model to mouse retinal neuronal data

#### Bilinear model reconstructs neuronal type-specific connectivity map from gene expression profiles

In our application of the bilinear model to the mouse retinal neuronal data, upon completion of the final training process, our optimized bilinear model produced transformation matrices, $\hat{\boldsymbol{A}}$ and $\hat{\boldsymbol{B}}$. We used these matrices to project the normalized single-cell transcriptomic data, $\hat{\boldsymbol{X}}$ and $\hat{\boldsymbol{Y}}$, into a shared latent feature space. Consequently, we obtained projected representations for BC and RGC types, $\hat{\boldsymbol{X}}\hat{\boldsymbol{A}}$ and $\hat{\boldsymbol{Y}}\hat{\boldsymbol{B}}$, respectively. With these latent representations, we were able to reconstruct the cell-type-specific connectivity matrix: $\hat{\boldsymbol{X}}\hat{\boldsymbol{A}}(\hat{\boldsymbol{Y}}\hat{\boldsymbol{B}})^T$ (*Figure 4a*).

To evaluate our model, we compared the reconstructed connectivity matrix with the one derived from connectomic data (*Figure 4b*). We calculated the Pearson correlation coefficient between entries of the two matrices to assess their agreement. The resulting correlation of 0.83 (p < 0.001)

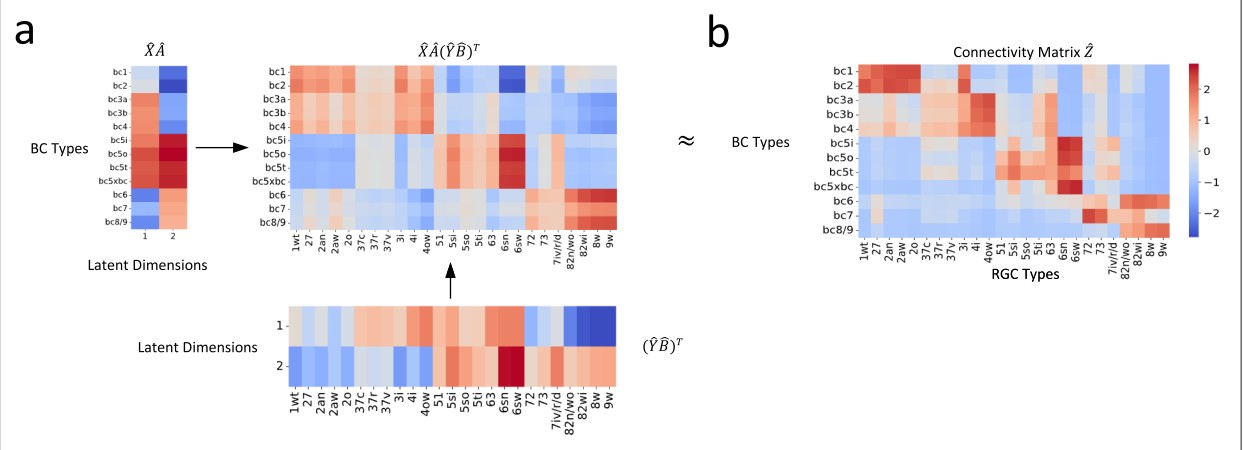

**Figure 4.** Reconstruction of connectivity map from gene expression profiles. (**a**) The reconstructed connectivity matrix, derived from the shared latent feature space projections. (**b**) The connectivity matrix obtained from connectomic data. Differences in color intensity represent the strength of connections, with dark red indicating strong connections and dark blue indicating weak or no connections.

The online version of this article includes the following figure supplement(s) for figure 4:

**Figure supplement 1.** Hyperparameter selection through cross-validation for the mouse retinal neuronal dataset.

**Figure supplement 2.** Heatmaps showcasing the average absolute cosine similarities across five optimization repetitions for (**a**) $\hat{A}$ and (**b**) $\hat{B}$.

**Figure supplement 3.** Detailed discrepancy analysis between the reconstructed and the target connectivity matrices.

demonstrated a robust association between the transformed gene expression features and the connectomic data. This result attests to our model's capability in capturing the relationship between these two distinct types of biological information.

To gain insights into our model's reconstruction accuracy, we employed the DS metric to identify entries with substantial deviations between the reconstructed and the actual connectivity matrices (*Figure 4—figure supplement 3a*; see 'Methods and supplementary materials' for details). This examination specifically quantified the extent of connections in the target matrix (positive entries) that were not captured in the model's reconstruction (negative entries; *Figure 4—figure supplement 3b and c*). Notably, the analysis revealed that only a small fraction, specifically 9 out of 115 connections, were not represented in the reconstructed matrix.

## Bilinear model recapitulates recognized connectivity motifs

Our cross-validation procedure indicated that the optimal number of latent dimensions was two (*Figure 4—figure supplement 1*; see 'Methods and supplementary materials' for details). This finding suggested that these two dimensions capture the essential connectivity motifs between BC and RGC types. This led us to further investigate what are these motifs and how they are different from each other.

We first reconstructed connectivity using only the first latent dimension. The first dimension appeared to emphasize connectivity patterns between BCs and RGCs that laminate within the IPL's central region, as well as those that laminate within the marginal region (*Figure 5a, d and g*). We then reconstructed connectivity using only the second latent dimension. Notably, the spotlight shifted to connections between BCs and RGCs that laminate within the outer and inner regions of the IPL, respectively (*Figure 5b, e and i*).

To confirm these observations, we further visualized BC and RGC types within the two-dimensional latent feature space (*Figure 5c and f*). Grouping BC and RGC types based on whether they fell within the positive or negative halves of the latent dimensions, we color-coded their stratification profiles within the IPL by group. BCs and RGCs that fell within the positive half of latent dimension 1 tend to stratify within the IPL's central region, delineated by the boundaries formed by the ON and OFF starburst amacrine cells (SACs; *Figure 5d and g*). Conversely, those falling within the negative half of this dimension tend to stratify in the marginal region of the IPL. As for the second latent dimension,

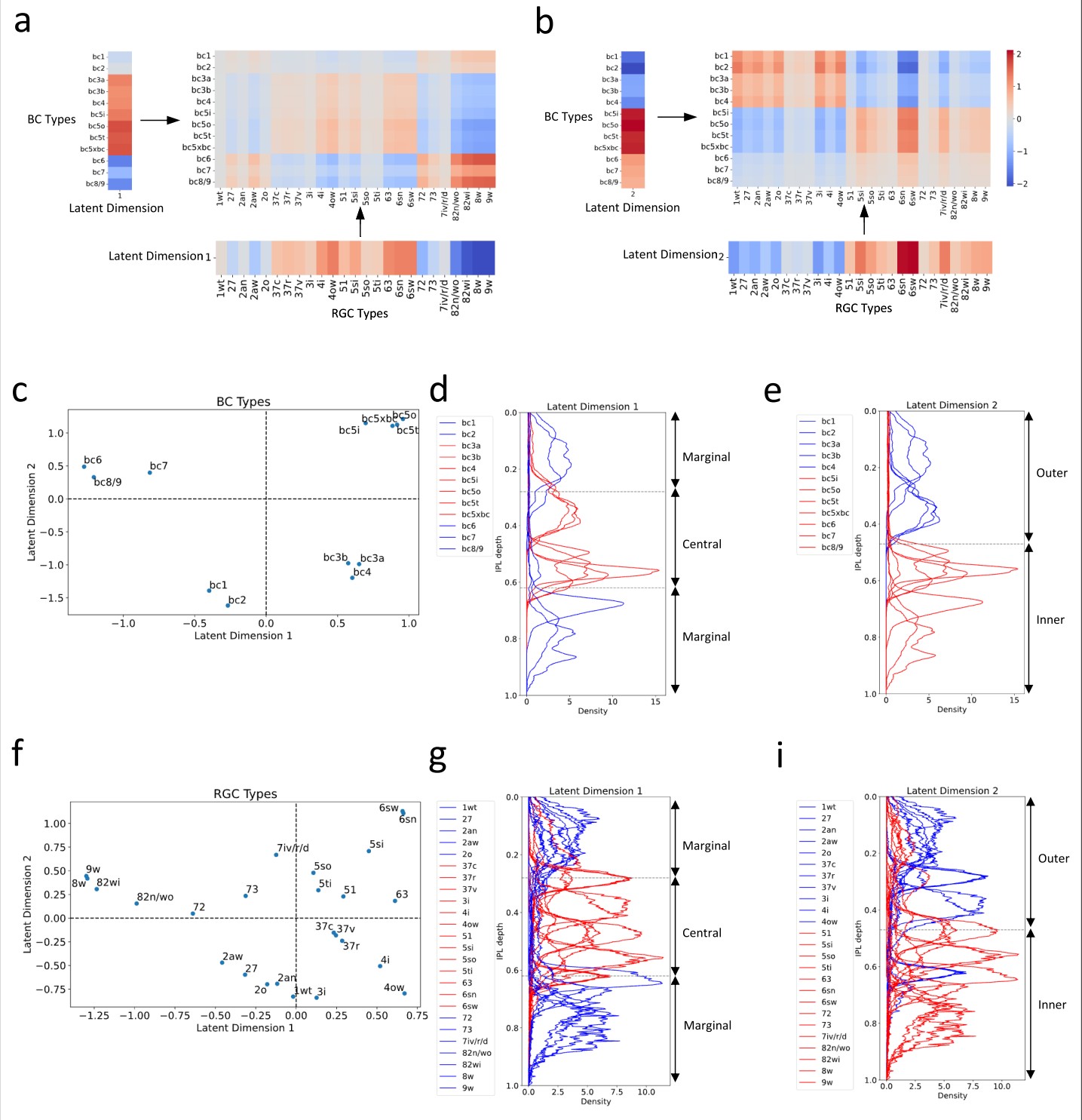

**Figure 5.** Distinct connectivity motifs revealed by the two latent dimensions. (**a, b**) The reconstructed connectivity using only latent dimension 1 or 2, respectively. Differences in color intensity represent the strength of connections. (**c**) BC types plotted in the latent feature space, with each point representing a specific BC type. Dashed lines indicate zero values for latent dimensions 1 and 2. (**d, e**) Stratification profiles of BC types in IPL, color-coded based on their positions along the first (**d**) or second (**e**) latent dimension. Red indicates BC types on the positive half, while blue indicates BC types on the negative half. (**f**) RGC types plotted in the latent feature space, with each point representing a specific RGC type. (**g, h**) Stratification profiles of RGC types in IPL, color-coded based on their positions along the first (**g**) or second (**h**) latent dimension. Dashed lines in (**d**) and (**g**) mark the positions of ON and OFF SACs (**Bae et al., 2018**). BCs and RGCs stratifying between them tend to exhibit more transient responses, and those stratifying outside them exhibit more sustained responses. Dashed lines in (**e**) and (**h**) denote the boundary of the outer and inner IPL (**Bae et al., 2018**). Synapses between BCs and RGCs in the outer retina mediate OFF responses, while those in the inner retina mediate ON responses.

BCs and RGCs that fell within the positive half predominantly stratify in the inner region of the IPL (*Figure 5e and i*), while those within the negative half primarily stratify in the IPL's outer region.

Interestingly, these distinct connectivity motifs align with two widely recognized properties of retinal neurons: kinetic attributes that reflect the temporal dynamics (transient versus sustained responses) of a neuron responding to visual stimuli, and polarity (ON versus OFF responses) reflecting whether a neuron responds to the initiation or cessation of a stimulus (*Euler et al., 2014*; *Sanes and Masland, 2015*; *Masland, 2012*; *Baden et al., 2016*). This correlation implies that our bilinear model has successfully captured key aspects of retinal circuitry from gene expression data.

## Bilinear model reveals interpretable insights into gene signatures associated with different connectivity motifs

The inherent linearity of our bilinear model affords a significant advantage: it enables the direct interpretation of gene expressions by examining their associated weights in the model. These weights signify the importance of each gene in determining the connectivity motifs between the BC and RGC types. We identified the top 50 genes with the largest positive or negative weights for BCs and RGCs across both latent dimensions. We plotted their weights alongside their expression profiles in the respective cell types (*Figure 6*).

Our analysis unveiled distinct gene signatures associated with the connectivity motifs revealed by the two latent dimensions. In the first latent dimension, genes like CDH11 and EPHA3, involved in cell adhesion and axon guidance, carried high weights for BCs forming synapses in the IPL's central region. In contrast, for BCs synapsing in the marginal region, we observed high weights in the cell adhesion molecule PCDH9 and the axon guidance cue UNC5D (*Figure 6a*). This pattern was echoed in RGCs but involved a slightly different set of molecules. For example, in RGCs forming synapses in the IPL's central region, the cell adhesion molecule PCDH7 carried high weights, whereas for RGCs synapsing in the marginal region, cell adhesion molecules PCDH11X and CDH12 were associated with high weights (*Figure 6b*).

The second latent dimension revealed a comparable pattern, albeit with different gene signatures. For BCs laminating in the IPL's outer region, high weights were assigned with guidance cues such as SLIT2, NLGN1, EPHA3, and PLXNA4, as well as the adhesion molecule DSCAM. For BCs in the inner region, the adhesion molecule CNTN5 was associated with a high weight (*Figure 6c*). In RGCs, we noticed that guidance molecules such as PLXNA2, SLITRK6, and PLXNA4 along with adhesion modules CDH8 and LRRC4C were associated with high weights for cells forming synapses in the IPL's outer region. In contrast, the adhesion molecule SDK2 was among the top genes for RGCs laminating and forming synapses in the IPL's inner region (*Figure 6d*). Some of these genes or gene families, such as Plexins (PLXNA2, PLXNA4), Contactin5 (CNTN5), Sidekick2 (SDK2), and Cadherins (CDH8,11,12), are known to play crucial roles in establishing specific synaptic connections (*Matsuoka et al., 2011*; *Sun et al., 2013*; *Peng et al., 2017*; *Krishnaswamy et al., 2015*; *Duan et al., 2014*; *Duan et al., 2018*; *Liu et al., 2018*). Others, particularly delta-protocadherins (PCDH7,9,11x), emerged as new candidates potentially mediating specific synaptic connections (*Sanes and Zipursky, 2020*).

To elucidate the biological implications of these identified gene sets, we further conducted Gene Ontology (GO) enrichment analysis on the top genes through g:Profiler, a public web server for GO enrichment analysis (*Reimand et al., 2007*; *Raudvere et al., 2019*). This tool allowed us to delve into the molecular functions, cellular pathways, and biological processes associated with these genes. Intriguingly, when we listed the top 10 significant GO terms for each latent dimension based on their adjusted p-values, we found two common themes: neuronal development and synaptic organization (*Supplementary file 4*). *Supplementary file 4* also highlights the number of the top genes associated with each GO term, revealing that overall about 47% of these genes are involved in neural development and synaptic organization. Such findings underscore the potential roles of these genes in forming and shaping the specific connections between BC and RGC types.

## Bilinear model predicts connectivity partners of transcriptionally defined RGC types

The success of recommendation systems in accurately predicting the preferences of new users inspired us to leverage the bilinear model for predicting the connectivity partners of RGC types whose interconnections with BC types remain uncharted. There are some RGC types defined from single-cell

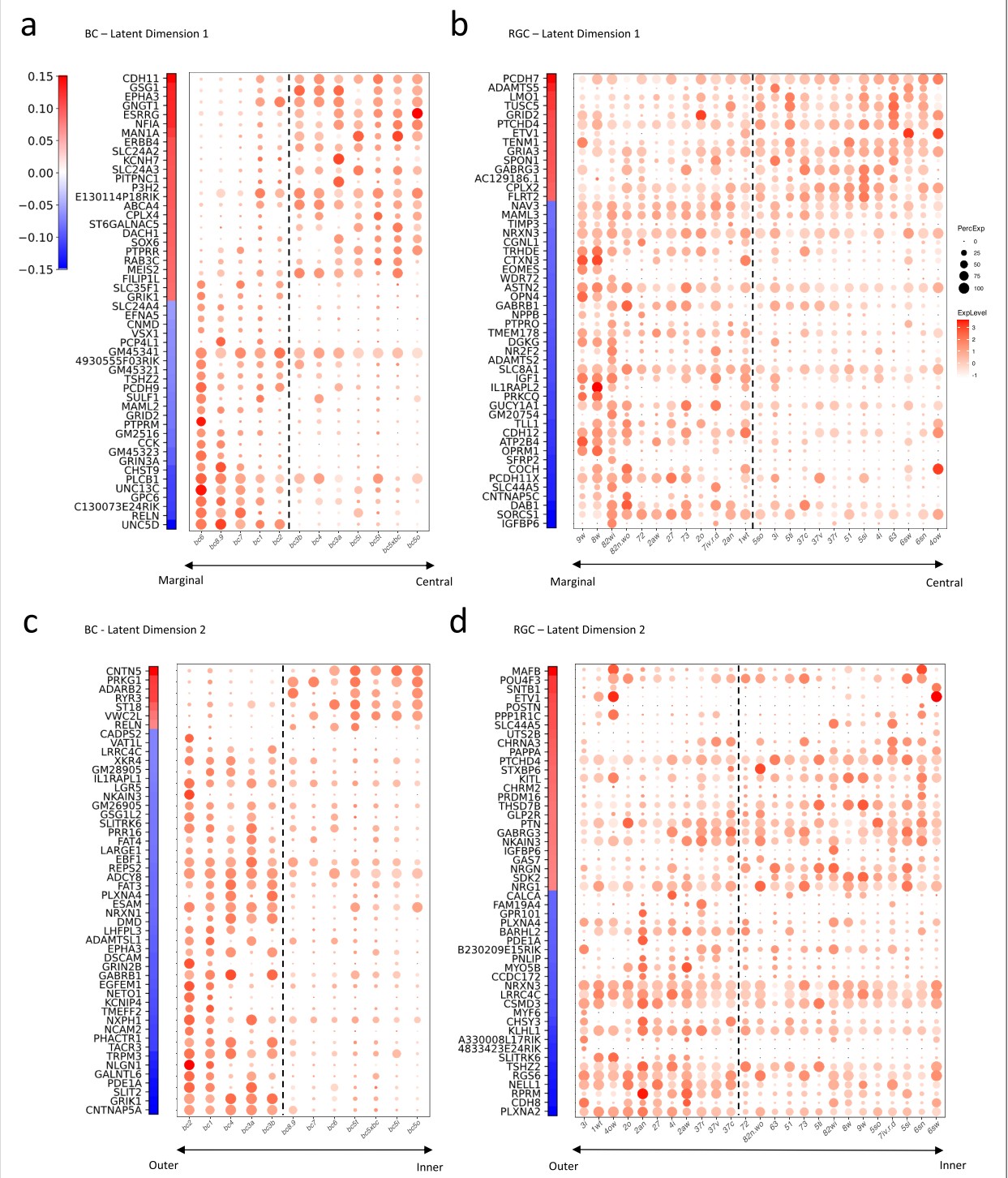

**Figure 6.** Gene signatures associated with the two latent dimensions. (**a, b**) Weight vectors of the top 50 genes for latent dimension 1, along with their expression patterns in BC types (**a**) and RGC types (**b**). The weight value is indicated in the color bar, with the sign represented by color (red: positive and blue: negative), and the magnitude by saturation. The expression pattern is represented by the size of each dot (indicating the percentage of cells expressing the gene) and the color saturation (representing the gene expression level). BC and RGC types are sorted by their positions along latent dimension 1, as shown in *Figure 5c and f*, with the dashed line separating the positive category from the negative category. (**c, d**) Weight vectors of the top 50 genes for latent dimension 2, and their expression patterns in BC types (**c**) and RGC types (**d**), depicted in the same manner as in (**a**) and (**b**). BC and RGC types are sorted by their positions along latent dimension 2.

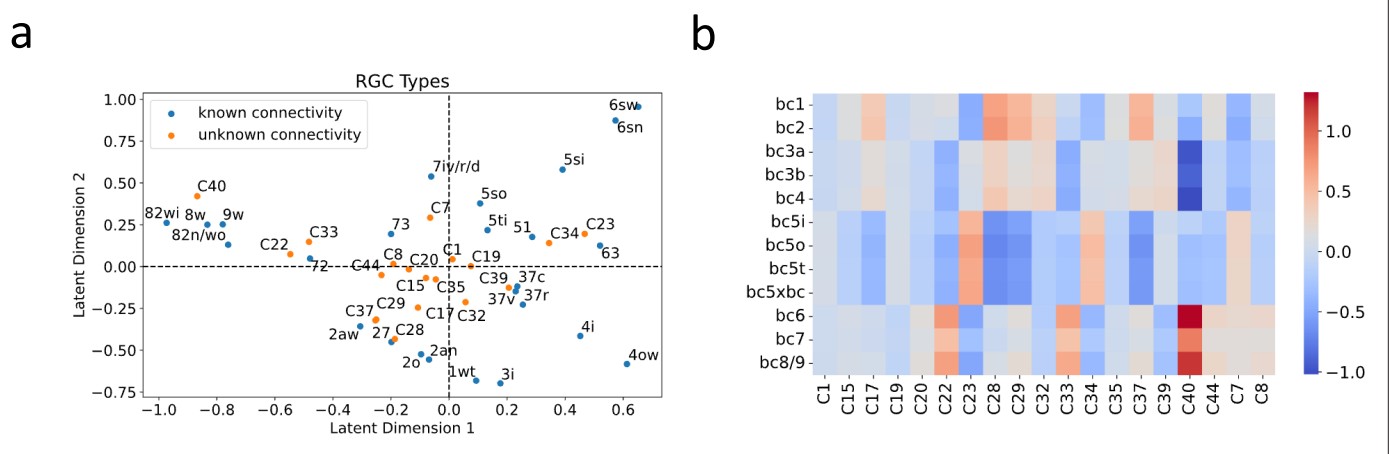

**Figure 7.** BC partner prediction of transcriptionally-defined RGC types. (**a**) Projection of transcriptionally-defined RGC types with unknown connectivity into the same latent space as those with known connectivity. (**b**) The resulting predicted connectivity matrix between these RGC types and BC types. Transcriptionally-defined RGC types are named according to *Tran et al., 2019*.

transcriptomic data (*Tran et al., 2019*), which lack clear correspondence with those identified through connectomics studies (*Bae et al., 2018*). This discrepancy leaves the connectivity patterns of these transcriptionally defined RGC types unknown, providing an opportunity for our model to predict their BC partners.

To accomplish this, we first projected these RGC types into the same latent space as those used to train the model (*Figure 7a*). We then employed this projection to construct a connectivity matrix between these RGC types and BC types (*Figure 7b*), facilitating educated estimates about their connectivity partners. For each transcriptionally defined RGC type, we identified the top three BC types as potential partners, determined by the highest values present in the connectivity matrix. These three BC types could provide insight into the potential synaptic input to each RGC type. Detailed predictions are presented in *Supplementary file 5*.

Although the ground truth connectivity of these RGC types remains unknown due to the absence of matching types in connectomic data, *Goetz et al., 2022*, via Patch-seq, attempted to match some transcriptomic types with functionally defined RGC types. These functional descriptions may hint at the BC partners of these RGC types. For instance, an RGC exhibiting OFF sustained responses is likely to be synaptically linked with BC types bc1-2, known to mediate OFF sustained pathways. Conversely, an RGC that displays ON sustained responses likely receives synaptic inputs from BC types bc6-9, which oversee ON sustained pathways. We summarized these functional descriptions in *Supplementary file 5*, referencing Figure 5A from *Goetz et al., 2022*, and highlighted whether our predictions were consistent with these functional annotations. Among the ten predictions made, eight aligned with these functional descriptions, lending support to the predictive power of our model.

## Discussion
### Summary of study
This study showcased a novel application of the bilinear modeling approach within the realm of gene expression analysis of neuronal type connectivity, drawing inspiration from recommendation systems - a machine learning domain focused on capturing intricate interactions between users and items and predicting user preferences. This analogy served as a useful framework in our study, where the roles of users and items in the recommendation systems are mirrored by presynaptic and postsynaptic neurons, respectively. Likewise, the user-item preference matrix corresponds to the synaptic connection matrix in neural circuits. The recommendation systems are based on the assumption that user preferences and item attributes can be represented by latent factors; similarly, our model assumes that synaptic connectivity between various neuron types is determined by a shared latent feature space derived from gene expression profiles.

The applicability and effectiveness of our bilinear model were validated using two different datasets. Applying it to the *C. elegans* neuronal dataset, which include data of gap junction connectivity and innexin expression at the individual neuron level, we showed that the model could be generalized to single-cell level connectivity by treating each neuronal type as an individual cell ('Gene expression and connectivity of each cell are known simultaneously'), and incorporate spatial constraints such as physical contact between neurons into the weight matrix ('Gap junction connectivity and innexin expression data of *C. elegans* neurons'). In a more complex scenario where the transcriptomic and connectomic data are from different sources and aligned at the neuronal-type level, we demonstrated the model's capability in decoding the genetic underlying of the connectivity between neuronal types ('Connectivity and gene expressions of neuronal types are from different sources'), using the mouse retinal neuronal dataset ('Single-cell transcriptomic and connectomic data of mouse retinal neurons'). This emphasizes the model's potential in offering insights into the genetic mechanisms that orchestrate synaptic connections across various nervous systems.

## Insights from analysis of *C. elegans* dataset and comparison with SCM

Using the *C. elegans* neuronal dataset, we conducted a comparative analysis between our bilinear model and the SCM, which correlates neuronal innexin expression with gap junction connectivity via a rule matrix (*Kovács et al., 2020*; *Barabási and Barabási, 2020*). The SCM incorporates spatial constraints, such as physical contact between neurons, and represents the connectome as an edge list for regression against the Kronecker product of the gene expression matrix. Our model is closely related to the SCM, as it can be seen as factorization of the rule matrix into the product of two lower-dimensional transformation matrices. This factorization not only yielded a performance comparable to, if slightly better than, the SCM in reconstructing the gap junction connectivity matrix, but also revealing potential new innexin interactions for experimental exploration (*Figure 2*; *Figure 3*).

Beyond these, a crucial advantage of our bilinear model lies in in its computational efficiency, an attribute of significance when scaling to larger datasets, where the number of genes and the number of neurons or neuronal types escalates to the order of thousands, such as those of the mouse or macaque cortex (*Yao et al., 2023a*; *Chen et al., 2023*). In such situation, the computational complexity of the SCM is substantial, given its reliance on the Kronecker product's dimensions and subsequent matrix inversion. In contrast, the computational demands of our bilinear model, driven primarily by matrix multiplication in gradient descent, are considerably more manageable, offering scalability and feasibility even as dataset sizes increase. Furthermore, the requirement to calculate the Kronecker product in SCM significantly heightens memory usage, critical when the data scale is large but memory resources are constrained. These advantages ensure our bilinear model a scalable solution when applied to other organisms and brain regions.

In assessing the bilinear model's and the SCM's performance to reconstruct *C. elegans* gap junction connectivity, the resulting modest ROC-AUC scores (approximately 0.64, much lower than the ideal 1.0) underscore the challenges in predicting electrical synapse specificity using innexin expressions alone. This suggests that additional molecular mechanisms, beyond innexin interactions, play crucial roles in forming specific electrical synaptic connections. Indeed, in the realm of chemical synapses, it's increasingly recognized that synaptic specificity is significantly influenced by factors such as cell-cell adhesion and recognition molecules, rather than just the pre- or post-synaptic machinery (*Sanes and Zipursky, 2020*). Recent studies support this viewpoint. For instance, research on the *C. elegans* motor circuit has revealed how a developmental program fine-tunes cAMP signaling to guide neuron-specific assembly of electrical synapses (*Palumbos, 2021*). Furthermore, the observed coexistence of electrical and chemical synapses in close proximity intimates potential shared mechanisms underlying their specificity (*Lasseigne et al., 2021*).

## Insights from application to mouse retinal neuronal dataset

Applying to the mouse retinal neuronal dataset, our bilinear model successfully reconstructed a neuronal type-specific connectivity map from gene expression profiles and recapitulated two core connectivity motifs of the retinal circuit, representing synapses formed in central or marginal parts of the IPL, and synapses formed in outer or inner regions (*Figure 4*; *Figure 5*). These motifs align well with recognized properties of retinal neurons: kinetic attributes (transient versus sustained responses) and polarity (ON versus OFF responses; *Euler et al., 2014*; *Sanes and Masland, 2015*; *Masland*,

2012; *Baden et al., 2016*). Significantly, these motifs aren't predefined or explicitly encoded into the model; instead, they emerge naturally from the model, further attesting to the model's power to capture key aspects of retinal circuitry.

The bilinear model also revealed unique insights into the gene signatures associated with the connectivity motifs. The weight vectors in the transformation matrices provide a means to assess the relative importance of individual genes. This direct interpretability is a significant advantage of the linear model, allowing for a more intuitive understanding of the gene-to-connectivity transformation process. Our analysis discovered distinct gene signatures associated with different connectivity motifs (*Figure 6*). Among these genes, some have been previously implicated in mediating specific synaptic connections, thererby validating our approach. For instance, Plexins A4 and A2 (PLXNA4, PLXNA2), predicted to be crucial for RGCs' synapsing in the outer IPL, have been shown to be necessary for forming specific lamina of the IPL in the mouse retina, interacting with the guidance molecule Semaphorin 6 A (SEM6A) (*Matsuoka et al., 2011*; *Sun et al., 2013*). Contactin5 (CNTN5), which our model predicted as vital for BCs forming synapses in the inner IPL, has been shown to be essential for synapses between ON BCs and the ON lamina of ON-OFF direction-selective ganglion cells (ooDSGCs; *Peng et al., 2017*). Sidekick2 (SDK2), predicted to be critical for RGCs' synapses in the inner IPL, has been shown to guide the formation of a retinal circuit that detects differential motion (*Krishnaswamy et al., 2015*). Similarly, Cadherins (CDH8,11,12), whose combinations have been implicated in synaptic specificity within retinal circuits (*Duan et al., 2014*; *Duan et al., 2018*), were highlighted for multiple connectivity motifs. In particular, Cadherin8 (CDH8), which our model predicted to be crucial for RGC's synaptic connections in the outer IPL, has been shown to be guided by the transcriptional factor Tbr1 for laminar patterning of J-RGCs, a type of OFF direction-selective RGCs (*Liu et al., 2018*). In addition to these validated gene signatures, our analysis identified promising candidate genes that may mediate specific synaptic connections. Particularly, delta-protocadherins (PCDH7,9,11x) appeared as potential new candidates. While their roles in synaptic connectivity aren't fully understood (*Sanes and Zipursky, 2020*), mutations in delta-protocadherins in mice and humans have been linked with various neurological phenotypes, including axon growth and guidance impairments and changes in synaptic plasticity and stability (*Kahr et al., 2013*; *Light and Jontes, 2017*; *Peek et al., 2017*; *Bisogni et al., 2018*). Future experimental studies are needed to validate these findings and further unravel the roles of these genes in neural circuit formation and function in the mouse retina.

The bilinear model's utility extends beyond the identification of gene signatures, emerging as a potent tool for hypothesis generation, particularly in predicting connectivity for transcriptionally defined neuronal types whose synaptic partners remain uncharted (*Figure 7*). Trained on data from a specific neural region, the bilinear model can facilitate the anticipation of synaptic partners for newly characterized transcriptional types within that region, thereby generating hypotheses on their functional roles within neural circuits. Furthermore, this model opens avenues for inferring neural wiring alterations resulting from genetic manipulations. For instance, by altering the genetic profile of certain neuronal types to create new transcriptionally defined types, we can use the model to predict changes in their synaptic partners, offering insights into the consequent reconfiguration of neural networks. This could be further extended to hypothesize the rewiring of the brain under psychological disorders, such as autism, where significant connectome changes suggest shifts in synaptic partner choices (*Roine et al., 2015*; *Hong et al., 2019*). With recent availability of neuronal gene expression data of autism (*Velmeshev et al., 2019*; *Nassir et al., 2021*; *Li et al., 2023*), our model stands poised to predict the implications of such genetic profiles on neural circuitry, guiding the research of understanding and treating this psychological disorder.

While our bilinear model offers valuable insights into the connectivity motifs of retinal circuits and the associated gene signatures, with many findings aligning with existing literature, it is important to acknowledge certain limitations of this study. Firstly, the model's connectivity matrix was deduced from stratification profiles derived from EM reconstruction. Although prior research has indicated stratification as a meaningful indicator of connectivity within the mouse retina, as certain BC types preferentially connect with specific RGC types stratified in the same lamina (*Duan et al., 2014*; *Krishnaswamy et al., 2015*; *Duan et al., 2018*), this metric may not capture the entire complexity of synaptic connections (*Dunn and Wong, 2014*). The incorporation of additional experimental data, such as electrophysiological measurements, could enhance both the accuracy and the reliability of the connectivity metrics. Secondly, the model, despite its overall success in reconstructing the connectivity

matrix, missed several connections, notably among specific BC-RGC pairs such as those between RGC types 51, 5ti and BC types 3a, 3b, and 4 (*Figure 4—figure supplement 3*). This highlights the potential for a more complex approach, such as deep learning models, to capture the subtleties of synaptic connections. Finally, the list of top genes identified by our model is enriched with genes directly mediating synapse formation and maintenance, such as adhesion molecules (*Figure 6*; *Supplementary file 4*), yet overlooks transcription factors like Tbr1 known to affect synaptic specificity (*Liu et al., 2018*). These factors, impacting various neuronal functionalities, might not be captured by a linear model that inherently favors predictor variables that strongly correlate with the target variable.

## Future directions

### Experiment validation of candidate genes

The bilinear model enables the predictions of possible changes in synaptic connections resulting from changes in expressions of the candidate genes. Emerging genome editing technologies, particularly CRISPR/Cas9 (*Cong et al., 2013*; *Mali et al., 2013*), offers a precise and efficient way to validate these predictions through experiments. By leveraging CRISPR/Cas9, targeted genetic manipulations, such as gene silencing or modification, can be conducted to assess their impact on synaptic connectivity. In the context of the mouse retina, the delivery of CRISPR/Cas9 components into BCs or RGCs can be achieved through electroporation or adeno-associated virus (AAV) vectors, respectively, allowing for targeted gene intervention (*Sarin et al., 2018*; *Tian et al., 2022*).

The finding of delta-protocadherins (PCDH7,9,11x) as potential mediators of synaptic specificity in the mouse retina presents an exciting opportunity for experimental exploration. We propose to design CRISPR/Cas9 systems targeting these delta-protocadherins (PCDH7,9,11x), similar to those detailed in a recent study (*Biswas et al., 2021*). Delivered to the mouse retina using AAV vectors, we expect to knockdown delta-protocadherin expressions in RGCs (*Tian et al., 2022*). With PCDH7 identified as a key factor in synapse formation within the central regions of the IPL, a focal point of our investigation will be RGC types like W3B RGCs, which are known to stratify in these central layers (*Zhang et al., 2012*). The consequences of PCDH7 downregulation on the connectivity of W3B RGCs can be examined through multiple approaches (*Krishnaswamy et al., 2015*): immunohistochemical techniques or the use of transgenic markers can reveal morphological changes indicative of altered connectivity; electrophysiological assessments, such as targeted recordings from postsynaptic neurons while optogenetically stimulating presynaptic partners, offer a functional probe into the synaptic alterations. Similarly, as PCDH9 and PCDH11x are implicated in synaptic connections within the marginal regions of the IPL, candidate RGCs for examination could include ON and OFF sustained alpha RGCs, known for their peripheral stratifications (*Krieger et al., 2017*).

This experimental paradigm is not confined to the mouse retina but extends to a broad range of neuronal circuits, thanks to the flexibility and wide applicability of genome editing tools like CRISPR/Cas9 (*Dickinson and Goldstein, 2016*; *Gratz et al., 2015*; *Li et al., 2016*). The capacity to induce targeted gene knockouts or modifications will empower researchers to validate our bilinear model's predictions and explore the underlying genetic mechanisms for synaptic formation and maintenance. This endeavor opens new avenues for deciphering the complex interplay between genetics and neural circuit wiring, furthering our comprehension of the molecular mechanisms driving synaptic specificity.

### Application to other neural systems

Our bilinear model, while illustrated using the *C. elegans* and mouse retina datasets, holds significant potential for elucidating the genetic underpinnings of neuronal connectivity across various species and brain regions, contingent upon the availability of comprehensive gene expression profiles and synaptic connection data. For instance, the advent of a comprehensive single-cell transcriptome atlas for the adult fruit fly brain, alongside the recent establishment of its complete connectome, offers a fertile ground for extending our model to decipher the complex neural circuits of *Drosophila* (*Davie et al., 2018*; *Ding et al., 2023*).

In the context of the mouse brain, the depth and breadth of single-cell sequencing efforts have unveiled a rich tapestry of transcriptomic cell types across cortex regions and the hippocampus (*Tasic et al., 2016*; *Tasic et al., 2018*; *Yao et al., 2021*; *Yao et al., 2023a*). These efforts, in tandem with connectomic studies that meticulously map neuronal connections, lay a foundation for integrating transcriptomic and connectomic data (*Bock et al., 2011*; *Lee et al., 2016*; *Turner et al., 2022*; *Yao*

*et al., 2023b*). Such integration, especially across diverse brain regions, presents an exciting avenue to uncover both neuronal connection mechanisms that are shared by neuronal types across different regions and those unique to specific regions. The scalability of our bilinear model, akin to collaborative filtering's effectiveness in e-commerce domains, supports the prospect of its cross-regional application. This approach positions our model at the forefront of efforts to explore how gene expression patterns contribute to the diversity of neuronal circuits across brain areas, moving us closer to a holistic understanding of the genetic blueprint of neuronal connectivity throughout the entire brain.

Nevertheless, we recognize the challenge that such well-aligned connectomic and transcriptomic data may not always be readily available. To address this, future research endeavors will also explore adaptations of our model to other available datasets, such as those that combine single-cell transcriptomic profiling with long-range neuronal projection mapping (*Chen et al., 2019*; *Sun et al., 2021*). Furthermore, our model is amenable to integration with trans-synaptic tracer-based sequencing methods (*Tsai et al., 2022*; *Zhang et al., 2023*), expanding its utility in studies where detailed connectomic information is limited. Pursuing these avenues is pivotal in broadening the model's utility and ensuring its relevance across a wider spectrum of brain connectivity research, making it an invaluable tool in the quest to unravel the complexities of neural circuitry.

## Model advancements

To enhance the model's fidelity and applicability, we propose several advancements. First, we recommend the integration of auxiliary data types, including electrophysiological data, neuron tracing data, and an array of omics data such as proteomics and epigenetics data, to augment and enrich the model's training base (*Baden et al., 2016*; *Tsai et al., 2022*; *Zhang et al., 2023*; *Mazan-Mamczarz et al., 2022*; *Bennett et al., 2023*). These data modalities offer complementary insights into neuronal function and connectivity, providing valuable context that can inform and refine the model's predictions.

Second, we envision extending the bilinear model to incorporate non-linear interactions, capturing the intricate dynamics between gene expressions and synaptic connections. A potential pathway for this is through kernel methods or the integration of neural networks, specifically adopting the 'two-tower model' framework renowned in modern recommendation systems (*Figure 8*). In this model, each 'tower' is a deep neural network that undertakes the non-linear transformation of input features

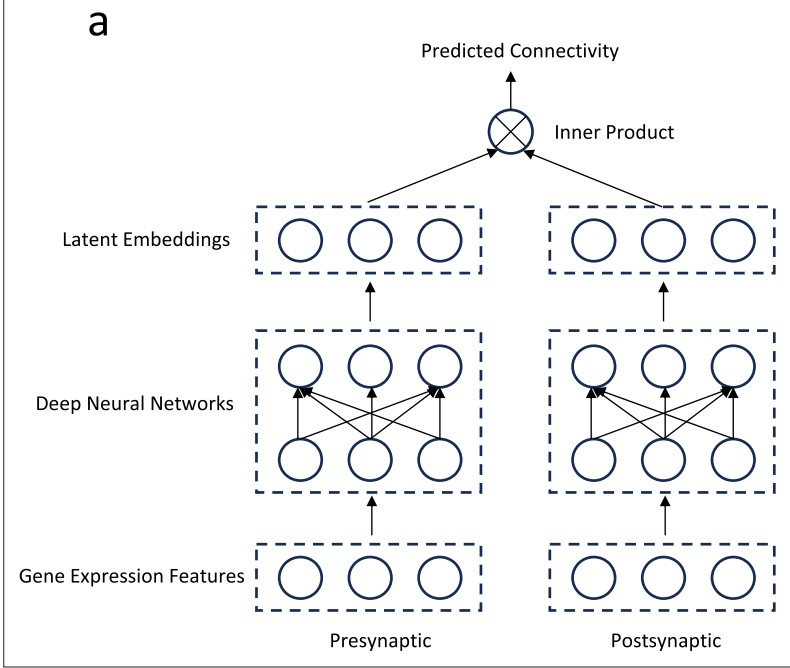

**Figure 8.** Future direction: A two-tower deep learning model. (**a**) Gene expression profiles of pre- and post-synaptic neurons are transformed into latent embedding representations via deep neural networks. The connectivity metric between the pre- and post-synaptic neurons is predicted by taking the inner product of their respective latent embeddings.

(*Wang et al., 2021*; *Yu et al., 2021*). This architecture has proven effective in capturing complex user-item interactions and could significantly enhance our model's ability to decipher the nuanced relationships between genetics and neural connectivity.

## Materials and methods
### Datasets and pre-processing

To validate and assess the efficacy of our bilinear model, we utilized two distinct datasets available from previous studies:

### Gap junction connectivity and innexin expression data of *C. elegans* neurons

We first used a dataset of gap junction connectivity and innexin expressions of individual *C. elegans* neurons. Derived from the work of *Cook et al., 2019* and subsequently analyzed by *Kovács et al., 2020*, this dataset included expression profiles of 18 innexin genes across 184 neurons, alongside detailed gap junction connectivity between these neurons. We followed the same procedure outlined by Kovács et al. to obtain the innexin expression matrix $X$ and $Y$ (in this case $X = Y$ with the dimensions of 184 × 18), and the connectivity matrix between individual *C. elegans* neurons $Z$.

To incorporate spatial constraints by considering only neuron pairs in physical contact, we extracted a contact matrix from the dataset. This was transcribed into the weight matrix $W$ in our model, with values set to 0 for neuron pairs without physical contact and 1 for those with contact. This enabled our bilinear model to focus on the 5,592 neuron pairs that exhibit physical contacts, restricting the analysis to biologically plausible connections.

The utilization of this dataset serves a dual purpose. It not only provides a validation for our bilinear model but also enables a direct comparison with the model employed by Kovács et al., offering a comprehensive evaluation of the bilinear model in the context of established connectomic research.

### Single-cell transcriptomic and connectomic data of mouse retinal neurons

The second dataset encompassed data of mouse retinal neurons, integrating single-cell transcriptomic data from various studies with connectomic data obtained from the EyeWire project. The data provide us with connectivity information and gene expression profiles of mouse BCs and RGCs, and are important for applying our proposed bilinear model and testing its effectiveness in a more complex neuronal environment compared to the *C. elegans* dataset.

The single-cell transcriptomic data include the gene expression profiles for two classes of mouse retinal neurons – presynaptic BCs as reported by *Shekhar et al., 2016*, and postsynaptic RGCs as reported by *Tran et al., 2019*.

Preprocessing of this data adhered to previously documented procedures (*Shekhar et al., 2016*; *Tran et al., 2019*; *Qiao, 2023*). The transcript counts within each cell were first normalized to align with the median number of the transcripts per cell, followed by a log-transformation of the normalized counts. High variable genes (HVGs) were then selected using an approach based on establishing a relationship between mean expression level and the coefficient of variance (*Chen et al., 2016*; *Pandey et al., 2018*; *Kurmangaliyev et al., 2019*). We focused on those cells whose types correspond with the neuronal types outlined in the connectomic data, as delineated later in *Supplementary file 1*, *Supplementary file 2*, and *Supplementary file 3*. This yielded two matrices, $X$ and $Y$, representing presynaptic BCs and postsynaptic RGCs, where each row pertains to a cell and each column represents an HVG. The dimensions of $X$ and $Y$ are 22453 × 17144 and 3779 × 12926, respectively.

Next, we performed a principal component analysis (PCA) on these matrices to transform the gene expression data into the principal component (PC) space. We retained only the PCs that account for a cumulative 95% of explained variance. Consequently, the gene expression of the BCs in $X$ and the RGCs in $Y$ were featurized by their respective PCs, resulting in matrices of dimensions 22453 × 11323 and 3779 × 3142, respectively.

Based on each cell's neuronal type, we computed the variance of gene expression features within these types. Mathematically, the variance of gene expression feature $m$ within the BC types and the RGC types are expressed as:

$$\hat{\sigma}_{x_{im}} = \sum_{i=1}^{a} \left( \frac{1}{n_i} \sum_{k=1}^{n_i} (x_{(ik)m} - x_{(i.)m})^2 \right) \tag{15}$$

$$\hat{\sigma}_{y_{jm}} = \sum_{j=1}^{b} \left( \frac{1}{n_j} \sum_{l=1}^{n_j} (y_{(jl)m} - y_{(j.)m})^2 \right) \tag{16}$$

Taking $\bar{x}_{im}$ and $\bar{y}_{jm}$ to represent the average gene expression feature $m$ of the BC type $i$ and the RGC type $j$, we were able construct matrices, $\hat{X}$ and $\hat{Y}$, in which $\hat{x}_{im} = \frac{\bar{x}_{im}}{\hat{\sigma}_{x_m}}$ and $\hat{y}_{im} = \frac{\bar{y}_{im}}{\hat{\sigma}_{y_m}}$. In these matrics, each row represents a cell type, with the dimensions of $\hat{X}$ being $25 \times 11323$ and $\hat{Y}$ being $12 \times 3142$. These matrices serve to bridge the gene expression of BC types and RGC types with the connectivity matrix of these neuronal types derived from the connectomic data.

The connectivity matrix of neuronal types is derived from connectomic data acquired through the process of serial electron microscopy (EM)-based reconstruction of brain tissues (**Denk and Horstmann, 2004**; **Helmstaedter et al., 2013**; **Tapia et al., 2012**). From these reconstructed tissues, connectivity measurements are usually expressed as either the contact area or the number of synapses between neurons (**Helmstaedter et al., 2013**; **Turner et al., 2022**). When normalized to the total contact area or total number of synapses of each neuron, the resulting metric, ranging from 0 to 1, signifies the percentage of contact area or synapses formed between neurons. This normalized metric provides a quantitative connectivity measure, where 0 indicates no connectivity and 1 implies complete connectivity between two neurons.

Our analysis utilized the neural reconstruction data of mouse retinal neurons, courtesy of the EyeWire project, a crowd-sourced initiative that generates 3D reconstructions of neurons from serial section EM images (**Kim et al., 2014**). This extensive dataset facilitated the derivation of a comprehensive connectivity matrix between two classes of mouse retinal neurons - BCs (**Greene et al., 2016**) and RGCs (**Bae et al., 2018**). The data were sourced from the EyeWire Museum (https://museum.eyewire.org/), which offers detailed information for each cell in a JSON file, including attributes like 'cell id', 'cell type', 'cell class', and 'stratification'. The stratification profile describes the linear density of voxel volume as a function of the inner plexiform layer (IPL) depth (**Kim et al., 2014**; **Greene et al., 2016**; **Bae et al., 2018**).

We approximated the connectivity metric between a BC and a RGC using the cosine similarity of their stratification profiles. Let $v_{ik}$ and $v_{jl}$ denote the stratification profiles of the $k^{th}$ cell in BC type $i$ and the $l^{th}$ cell in RGC type $j$, respectively. The connectivity metric $z_{(ik)(jl)}$ between these two neurons can be expressed as:

$$z_{(ik)(jl)} = \frac{\boldsymbol{u}_{ik}\boldsymbol{v}_{jl}}{|\boldsymbol{v}_{ik}||\boldsymbol{v}_{jl}|} \tag{17}$$

This equation represents the degree of overlap in their voxel volume profile within the IPL, resulting in the connectivity matrix $\boldsymbol{Z}$ between mouse BCs and RGCs. To allow for both positive and negative values within the matrix, we standardized $\bar{\boldsymbol{Z}}$ by subtracting the mean of $\bar{\boldsymbol{Z}}$ and then dividing by its standard deviation. Subsequently, the connectivity matrix $\bar{\boldsymbol{Z}}$ between mouse BC and RGC neuronal types was calculated, with each element $\bar{z}_{ij} = z(i.)(j.)$ representing the average of the connectivity metrics between cells of BC type $i$ and cells of RGC type $j$.

Aligning neuronal types as annotated in the single-cell transcriptomic data and those identified in the connectomic data was informed by findings from previous studies. Notably, a one-to-one correspondence exists between BC cell types classified by **Shekhar et al., 2016** and **Greene et al., 2016**. This correspondence is presented in **Supplementary file 1**.

Regarding RGC types, alignment between cell types annotated in **Tran et al., 2019** and **Bae et al., 2018** was established primarily based on the findings from **Goetz et al., 2022**. This study presents a unified classification of mouse RGC types, based on their functional, morphological, and gene expression features. The corresponding RGC types were mainly obtained from Supplementary Table S3 of **Goetz et al., 2022** (**Supplementary file 2**), with additions derived from Supplementary Table S1 of **Tran et al., 2019** based on the expressions of genetic markers of these RGC types (**Supplementary file 3**).

## Model training, validation, and comparison

Our approach of training and validating the bilinear model involved an iterative optimization of transformation matrices using the AGD algorithm, as outlined in 'Bilinear model for neuronal type connectivity'. The primary goal was to minimize the defined loss function. With the matrices initially generated from a standard normal distribution, the optimization process continued until the loss change was less than a threshold of $10^{-6}$, or a maximum of $10^6$ iterations were completed.

During optimization, we focused on two key hyperparameters: the regularization parameters, $\lambda_A$ and $\lambda_B$, and the latent feature space dimensionality. Preliminary tests indicated that a lower loss was achieved when both regularization parameters were set equally, leading us to consolidate them into a single parameter, $\lambda$.

### *C. elegans* neuronal dataset

For the *C. elegans* dataset, which provides simultaneous gene expression and connectivity data for individual cells, we employed the model configuration described in 'Gene expression and connectivity of each cell are known simultaneously'. The model's hyperparameters, $\lambda$ and the latent feature space dimensionality, were fine-tuned through five-fold cross-validation, exploring a range of values for $\lambda$ and different dimensions for the latent feature space. The optimal hyperparameters were identified based on the lowest validation loss observed during cross-validation (*Figure 2—figure supplement 1*).

Given the prior utilization of this dataset in validating the SCM proposed by *Kovács et al., 2020*, our bilinear model was positioned for a direct comparison with the SCM. The SCM introduced a rule matrix $O$ with the aim to minimize the discrepancy between the observed connectivity and the gene expression-based predicted connectivity $XOX^T$, employing L2 regularization on $O$. Our bilinear model echoes this approach, where we seek to minimize the divergence between the connectivity matrix and the bilinearly predicted connectivity $XA(XB)^T$, with L2 regularization imposed on matrices $A$ and $B$. In essence, the bilinear form decomposes the rule matrix into two lower-dimensional matrices, which represent projections onto latent dimensions.

To quantitatively compare the bilinear model's transformation matrix product $\hat{O} = AB^T$ with the SCM's rule matrix $O$, and to systematically identify the genetic interaction each model uniquely captured, we introduced the discrepancy score (DS). For each pair of corresponding entries in the matrices at indices $i$ and $j$, the DS is calculated as follows:

$$DS_{ij} = \frac{|\hat{o}_{ij} - o_{ij}|}{|\hat{o}_{ij}| + |o_{ij}|} \tag{18}$$

This metric, ranging from 0 to 1, quantifies the relative discrepancy between the two matrices, normalizing it in relation to their magnitudes. A score close to 1 indicates a large discrepancy, while a score near 0 suggests a negligible difference between the entries. Through this lens, we can further scrutinize the corresponding entries with the score above a certain threshold to reveal specific genetic interactions captured by one model but potentially missed by the other.

### Mouse retinal neuronal dataset

The model's application to the mouse retina dataset, which involves gene expression and connectivity data from disparate sources, was facilitated by the approach outlined in 'Connectivity and gene expressions of neuronal types are from different sources'. Optimal hyperparameters were determined through five-fold cross-validation, adjusting $\lambda$ and exploring various dimensionalities for the latent feature space (*Figure 4—figure supplement 1*). Notably, the lowest validation loss was achieved with the dimensionality of two. Given the chosen hyperparameters, we performed the final round of training on the entire dataset to yield the definitive transformation matrices $\hat{A}$ and $\hat{B}$.

To assess the consistency of our model under PCA pre-processing across different replicates, we repeated the optimization procedure five times, each time adhering to the previously identified optimal hyperparameters. In the context of our solution, where $\hat{A} = \begin{bmatrix} u_1 & u_2 \end{bmatrix}$ and $\hat{B} = \begin{bmatrix} v_1 & v_2 \end{bmatrix}$, with vectors $u_1, v_1$ representing coefficients for the first latent dimension and $u_2, v_2$ for the second, we noted that negating the coefficients of any latent dimension in both matrices (for instance, $\hat{A} = \begin{bmatrix} -u_1 & u_2 \end{bmatrix}$ and $\hat{B} = \begin{bmatrix} -v_1 & v_2 \end{bmatrix}$) results in an equivalent solution. Therefore, to compare solutions across different

repetitions, we calculated the absolute value of cosine similarity for each latent dimension's coefficient vectors, and reported the similarity between solutions as the average of the values across the two latent dimensions. Moreover, we recognized that swapping the positions of the coefficient vectors (yielding $\hat{A} = \begin{bmatrix} \boldsymbol{u}_2 & \boldsymbol{u}_1 \end{bmatrix}$ and $\hat{B} = \begin{bmatrix} \boldsymbol{v}_2 & \boldsymbol{v}_1 \end{bmatrix}$) also leads to an equivalent solution. To accommodate this, we evaluated both the original and swapped vector pairings for each repetition. The final measure of consistency was determined by taking the maximum of the two average absolute cosine similarities, ensuring a comprehensive and robust assessment of solution consistency across multiple runs.

We observed a high degree of consistency across multiple repetitions of the solutions under PCA pre-processing (*Figure 4—figure supplement 2*). The majority of the average absolute cosine similarity scores are close to 1, and even the minimum observed similarities are well above 0.75, suggesting that the optimization yields stable solutions.

## Additional information

### Funding
No external funding was received for this work.

### Author contributions
Mu Qiao, Conceptualization, Software, Formal analysis, Validation, Investigation, Visualization, Methodology, Writing - original draft, Writing - review and editing

### Author ORCIDs
Mu Qiao http://orcid.org/0000-0001-7309-4237

Reviewer #1 (Public Review): https://doi.org/10.7554/eLife.91532.3.sa1
Reviewer #2 (Public Review): https://doi.org/10.7554/eLife.91532.3.sa2
Author response https://doi.org/10.7554/eLife.91532.3.sa3

## Additional files

### Supplementary files
• Supplementary file 1. Correspondence of Mouse BC Types from *Greene et al., 2016*; *Shekhar et al., 2016*.

• Supplementary file 2. Correspondence of Mouse RGC Types from *Bae et al., 2018*; *Tran et al., 2019*; *Goetz et al., 2022*.

• Supplementary file 3. Additional Correspondence of Mouse RGC types (*Bae et al., 2018*; *Tran et al., 2019*; *Goetz et al., 2022*).

• Supplementary file 4. Gene Ontology (GO) Terms Associated with Latent Dimensions in BCs and RGCs.

• Supplementary file 5. Predicted BC Partners of Transciptionally-defined RGC Types.

• MDAR checklist

### Data availability
The current manuscript is a computational study, so no data have been generated for this manuscript. Modelling code is available at GitHub (copy archived at *Qiao, 2024*).

The following previously published datasets were used:

| Author(s) | Year | Dataset title | Dataset URL | Database and Identifier |
|---|---|---|---|---|
| Shekhar K, Lapan SW, Whitney IE, Tran NM, Macosko EZ, Kowalczyk M, Adiconis X, Levin JZ, Nemesh J, Goldman M, McCarroll SA, Cepko CL, Regev A, Sanes JR | 2016 | Single cell RNA-sequencing of retinal bipolar cells (Mouse BC Gene Expression Data) | https://www.ncbi.nlm.nih.gov/geo/query/acc.cgi?acc=GSE81905 | NCBI Gene Expression Omnibus, GSE81905 |
| Tran NM, Shekhar K, Whitney IE, Jacobi A, Benhar I, Hong G, Yan W, Adiconis X, Arnold ME, Lee JM, Levin JZ, Lin D, Wang C, Lieber CM, Regev A, He Z, Sanes JR | 2019 | Single-cell profiles of retinal neurons differing in resilience to injury reveal neuroprotective genes (Mouse RGC Gene Expression Data) | https://www.ncbi.nlm.nih.gov/geo/query/acc.cgi?acc=GSE137400 | NCBI Gene Expression Omnibus, GSE137400 |
| Kovács IA, Barabási DL, Barabás AL | 2020 | kpisti/SCM v1.0 (C. elegans Data) | https://zenodo.org/records/4027588 | Zenodo, 10.5281/zenodo.4027588 |

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
