## [Editor Report · eLife assessment]

This is an **important** computational study that applies the machine learning method of bilinear modeling to the problem of relating gene expression to connectivity. Specifically, the author attempts to use transcriptomic data from mouse retinal neurons to predict their known connectivity with promising results. On revision, the approach was tested against a second data set from *C. elegans*. A limited number of genes studied in this second dataset may have resulted in performance that matched but did not exceed prior models. However, taken together, the results were felt to provide **solid** evidence for the value of the approach.

---

## [Referee Report · Reviewer #1 (Public Review)]

Summary:

In this study, the author aimed to develop a method for estimating neuronal-type connectivity from transcriptomic gene expression data. They sought to develop an interpretable model that could be used to characterize the underlying genetic mechanisms of circuit assembly and connectivity in various neuronal systems.

Strengths:

Many of the proposed suggestions were addressed by the author from the initial review. In general the claims made by the author are more strongly supported by the data and better situated in the literature. A major improvement includes the application of the model to the *C. elegans* gap junction neuronal system. Despite several key differences in the dataset as compared to the mouse retina data, the proposed model performs comparably to the SCM model currently considered state of the art in the literature (the author should remain cautious about claiming better performance given extremely marginal differences). In section 7.2, the author clearly outlines additional advantages of the proposed model including superior time and space complexity. The overall model performance remains modest, but it learns the same rules as the SCM model as well as other candidate patterns.

As in the initial submission, the bilinear model recapitulates key connectivity motifs for the mouse dataset. The algorithm is shown to converge across several runs affirming its stability/replicability. The model is also extended to predict connectivity on unknown RGC-BC cell type pairs. Without ground truth, the author posits how it should perform based on known functional properties of the RGC type. The hypotheses are confirmed for 8/10 neuronal types with unknown connectivity. The author more clearly describes how this model can be used experimentally for hypothesis testing and presents a more comprehensive future roadmap regarding validation, avenues for improving the model, and incorporation of growing datasets.

Weaknesses:

While the C Elegans dataset is useful because it enables benchmarking to existing models, the dataset is quite different. The gene expression dimensionality is 18 genes as opposed to over 3000 genes in the mouse dataset. It is a strength that the model still works as intended, but a weakness that the bilinear model could not be tested on a similar mouse dataset. This distinction matters because it remains an open question if the PCA methodology would hold up in a dataset with varied distributions of gene expression. Variations of the PCA methodology could be evaluated further with the present dataset to make the generalizability of the model more convincing.

The Gene Ontology analysis requires more methodological explanation. The author claims, "(the linear nature of the model) enables the direct interpretation of gene expressions by examining their associated weights in the model. These weights signify the importance of each gene in determining the connectivity motifs between the BC and RGC types." If I am correctly understanding the methods, the model weights in each dimension are indexing the importance of a gene expression feature as opposed to the importance of a single gene alone, "the gene expression of the BCs in X and the RGCs in Y were featurized by their respective PCs, resulting in matrices of dimensions 22453 × 11323 and 3779 × 3142, respectively." It would be helpful to explain how gene weights are extracted from a gene expression feature once highlighted.

There could be a more rigorous analysis of the predictive capacity of the model even with the current data. The model recapitulates connectivity patterns from the full dataset and a prediction is demonstrated for unknown data. The model is thus championed as a useful tool for predicting how genetic modifications will influence connectivity, but this is not empirically evaluated.

Appraisal of whether the author achieved their aims, and whether results support their conclusions:

In line with the aims of the paper, the author proposed an interpretable bilinear model to learn a shared latent feature space derived from gene expression profiles to predict synaptic connectivity between various neuron types. The model was shown to generalize to two distinct neuronal systems with varying levels of genomic and cellular resolution. While the performance remains modest, the model performs comparably to the existing state of the art despite improved computational complexity.

Discussion of likely impact of the work on the field, and utility of methods and data to the community:

The author has elaborated substantially on the impact of this work, particularly how it could be leveraged in experimental settings. The clear methodology could be implemented by other researchers to test the model on new datasets and for benchmarking novel methods.

---

## [Referee Report · Reviewer #2 (Public Review)]

Summary:

In this study, Mu Qiao employs a bilinear modelling approach, commonly utilised in the recommendation systems, to explore the intricate neural connections between different pre- and post-synaptic neuronal types. This approach involves projecting single-cell Transcriptomic datasets of pre- and post-synaptic neuronal types into a latent space through transformation matrices. Subsequently, the cross-correlation between these projected latent spaces is employed to estimate neuronal connectivity. To facilitate the model training, Connectomic data is used to estimate the ground-truth connectivity map. This work introduces a promising model for the exploration of neuronal connectivity and its associated molecular determinants. In the revised version of the manuscript, the author has applied and validated the model in both *C. elegans* gap junction connectivity and the retina neuron connectivity conditions.

Strengths:

This study introduces a succinct yet promising computational model for investigating connections between neuronal types. The model, while straightforward, effectively integrates single-cell transcriptomic and connectomic data to produce a reasonably accurate connectivity map, particularly within the context of retinal connectivity. Furthermore, it successfully recapitulates connectivity patterns and helps uncover the genetic factors that underlie these connections.

Weaknesses:

(1) When compared with the previous method - SCM, the new model shows a similar performance level. This may be due to the limitation of the dataset itself, as it only has the innexin expression data. Is it possible to apply the SCM model to the more complete retina dataset and compare the performance with the proposed bilinear modelling approach?

Minor Weakness:

(1) The study lacks experimental validation of the model's prediction results.

---

## [Author Response]

The following is the authors’ response to the original reviews.

**eLife assessment**
This is a valuable computational study that applies the machine learning method of bilinear modeling to the problem of relating gene expression to connectivity. Specifically, the author attempts to use transcriptomic data from mouse retinal neurons to predict their known connectivity. The results are promising, although the reviewers felt that demonstration of the general applicability of the approach required testing it against a second data set. Hence the present results were felt to provide borderline incomplete support for a key premise of the paper.

We thank the reviewers for their insightful and constructive feedback. In response to the reviews, we have undertaken a comprehensive revision of our manuscript, incorporating changes and improvements as outlined below:

(1) New results have been included showcasing the application of our bilinear model to a seconddataset focusing on *C. elegans* gap junction connectivity. This extension validates our model with a biological context other than mouse retina and facilitates a direct comparison with the spatial connectome model (SCM).

(2) A new section titled "Previous Approaches" has been added to background, situating our studywithin the broader landscape of existing modeling methodologies.

(3) The discussion sections have been expanded to fully incorporate the suggestions and insightsoffered by the reviewers. This includes a deeper exploration of the implications of our findings, potential applications of our model, and a more thorough consideration of its limitations and future directions.

(4) To streamline the main text and ensure that the core narrative remains focused and accessible, select figures and tables have been relocated to the "Supplementary Materials" section.

**Reviewer 1 (Public Review):**
Summary of what the author was trying to achieve: In this study, the author aimed to develop a method for estimating neuronal-type connectivity from transcriptomic gene expression data, specifically from mouse retinal neurons. They sought to develop an interpretable model that could be used to characterize the underlying genetic mechanisms of circuit assembly and connectivity.Strengths:The proposed bilinear model draws inspiration from commonly implemented recommendation systems in the field of machine learning. The author presents the model clearly and addresses critical statistical limitations that may weaken the validity of the model such as multicollinearity and outliers. The author presents two formulations of the model for separate scenarios in which varying levels of data resolution are available. The author effectively references key work in the field when establishing assumptions that affect the underlying model and subsequent results. For example, correspondence between gene expression cell types and connectivity cell types from different references are clearly outlined in Tables 1-3. The model training and validation are sufficient and yield a relatively high correlation with the ground truth connectivity matrix. Seemingly valid biological assumptions are made throughout, however, some assumptions may reduce resolution (such as averaging over cell types), thus missing potentially important single-cell gene expression interactions.

Thank you for recognizing the strengths of our work, particularly the clarity of the model presentation and its foundation in recommendation systems. In the revised manuscript we have also extended the model’s capabilities to analyze gene interactions for neural connectivity at single-cell resolution, when gene expression and connectivity of each cell are known simultaneously.

Weaknesses:The main results of the study could benefit from replication in another dataset beyond mouse retinal neurons, to validate the proposed method. Dimensionality reduction significantly reduces the resolution of the model and the PCA methodology employed is largely non-deterministic. This may reduce the resolution and reproducibility of the model. It may be worth exploring how the PCA methodology of the model may affect results when replicating. Figure 5, ’Gene signatures associated with the two latent dimensions’, lacks some readability and related results could be outlined more clearly in the results section. There should be more discussion on weaknesses of the results e.g. quantification of what connectivity motifs were not captured and what gene signatures might have been missed.

We acknowledge the significance of validating our method across different datasets. In line with this, our revised manuscript now includes an expanded analysis utilizing a *C. elegans* gap junction connectivity dataset, which not only broadens the method’s demonstrated applicability but also underscores its versatility across varied neuronal systems.

To address the concern of resolution and reproducibility associated with PCA preprocessing, we have conducted a comparative analysis from five replicates of the bilinear model, presenting the results in the revised manuscript (Figure S3). This analysis confirms the consistency of the solutions, as evidenced by the similarity metrics. Furthermore, we discussed alternative methodologies, such as L1 or L2 regularization, to tackle multicollinearity, offering flexibility in preprocessing choices.

In response to feedback on the original Figure 5’s clarity, we have replaced the original Figure 5e-h with Table S4, which summarizes the gene ontology (GO) enrichment results and quantifies the number of genes associated with aspects of neural development and synaptic organization. This revision aims to improve the interpretability and accessibility of the results, ensuring a clearer presentation of the model’s insights.

Finally, we have expanded our discussion to address the study’s limitations more comprehensively. This includes exploration of potentially missed connections and gene signatures, such as transcription factors, which might not be captured by a linear model due to its inherent preference for predictors with strong correlations to the target variable.

The main weakness is the lack of comparison against other similar methods, e.g. methods presented in Barabási, Dániel L., and Albert-László Barabási. "A genetic model of the connectome." Neuron 105.3 (2020): 435-445. Kovács, István A., Dániel L. Barabási, and Albert-László Barabási. "Uncovering the genetic blueprint of the *C. elegans* nervous system." Proceedings of the National Academy of Sciences 117.52 (2020): 33570-33577. Taylor, Seth R., et al. "Molecular topography of an entire nervous system." Cell 184.16 (2021): 4329-4347.

We value your suggestion to compare our model with established methods. The revised manuscript now includes a comparative analysis with the spatial connectome model (SCM) using the same *C. elegans* dataset. In addition, a section reviewing previous approaches has been included in the background part, and the discussion part has been extended for the comparison.

Appraisal of whether the author achieved their aims, and whether results support their conclusions: The author achieved their aims by recapitulating key connectivity motifs from single-cell gene expression data in the mouse retina. Furthermore, the model setup allowed for insight into gene signatures and interactions, however could have benefited from a deeper evaluation of the accuracy of these signatures. The author claims the method sets a new benchmark for single-cell transcriptomic analysis of synaptic connections. This should be more rigorously proven. (I’m not sure I can speak on the novelty of the method)

In the revised manuscript. we emphasized the bilinear model’s innovative application in the context of neuronal connectivity analysis, inspired by collaborative filtering in recommendation systems. We present quantitative performance metrics, such as the ROC-AUC score and Pearson correlation coefficient, as well as its comparison with the SCM, to benchmark our model’s efficacy in reconstructing connectivity matrices. We also quantified the overlap of the genetic interactions revealed by the bilinear model and the SCM (using the *C. elegans* dataset), and reported the percentage of the top genes associated with neural development and synaptic organization (using the mouse retina dataset). These numbers set a precedent for future methodological comparisons.

Discussion of the likely impact of the work on the field, and the utility of methods and data to the community : This study provides an understandable bilinear model for decoding the genetic programming of neuronal type connectivity. The proposed model leaves the door open for further testing and comparison with alternative linear and/or non-linear models, such as neural networkbased models. In addition to more complex models, this model can be built on to include higher resolution data such as more gene expression dimensions, different types of connectivity measures, and additional omics data.

We are grateful for your recognition of the study’s potential impact. The bilinear model indeed offers a foundation for future explorations, allowing for integration with more complex models, higher-resolution data, and diverse connectivity measures.

**Reviewer 1 (Recommendations For The Authors):**
The inclusion of predicted connectivity (Figure 6) of unknown BC neurons is useful as it shows that this is a strong hypothesis generation tool. This utility should potentially be showcased more as it is also brought up in the abstract, "genetic manipulation of circuit wiring", with an explanation of how the model could be leveraged as such. The discussion may benefit from a summarizing sentence regarding which key gene signatures were identified and are in line with the literature, which key gene signatures/connectivity motifs may have been missed, and which gene signatures are novel.

Thank you for the insightful recommendation on emphasizing the model’s utility in generating hypotheses, particularly regarding predicting connectivity. In the revised manuscript, we have expanded the discussion on how our model can be leveraged to guide genetic manipulations at altering circuit wiring and highlighted its potential impact in the field.

We have discussed key gene signatures identified from our model that are in line with existing literature, such as plexins and cadherins, which have been previously recognized for their involvement in synaptic connection formation and maintenance. We have also introduced potential new candidates, such as delta-protocadherins. In the revised manuscript, we summarized potentially missed gene signatures or synaptic connections, to provide a comprehensive view of our findings.

**Reviewer 2 (Public Review):**
Summary:In this study, Mu Qiao employs a bilinear modeling approach, commonly utilized in recommendation systems, to explore the intricate neural connections between different pre- and post-synaptic neuronal types. This approach involves projecting single-cell transcriptomic datasets of pre- and post-synaptic neuronal types into a latent space through transformation matrices. Subsequently, the cross-correlation between these projected latent spaces is employed to estimate neuronal connectivity. To facilitate the model training, connectomic data is used to estimate the ground-truth connectivity map. This work introduces a promising model for the exploration of neuronal connectivity and its associated molecular determinants. However, it is important to note that the current model has only been tested with Bipolar Cell and Retinal Ganglion Cell data, and its applicability in more general neuronal connectivity scenarios remains to be demonstrated.Strengths:This study introduces a succinct yet promising computational model for investigating connections between neuronal types. The model, while straightforward, effectively integrates singlecell transcriptomic and connectomic data to produce a reasonably accurate connectivity map, particularly within the context of retinal connectivity. Furthermore, it successfully recapitulates connectivity patterns and helps uncover the genetic factors that underlie these connections.

Thank you for your positive assessment of the paper.

Weaknesses:(1) The study lacks experimental validation of the model’s prediction results.

We recognize the importance of experimental validation in substantiating the predictions made by computational models. While the primary focus of this study remains computational, we have dedicated a section in the revised manuscript, titled "Experimental Validation of Candidate Genes", to outline proposed methodologies for the empirical verification of our model’s predictions. This section specifically discusses the experimental exploration of novel candidate genes, such as deltaprotocadherins, within the mouse retina using AAV-mediated CRISPR/Cas9 genetic manipulation. We plan to collaborate with experimental laboratories to facilitate the validation. Given the extensive nature of experimental work, both in terms of time and resources, it is more pragmatic to present a comprehensive experimental investigation in a follow-up study.

(2) The model’s applicability in other neuronal connectivity settings has not been thoroughly explored.

The question of the model’s broader applicability is well-taken. In response, we have expanded our analysis to include additional neuronal data and connectivity settings. Specifically, the revised manuscript includes results where we apply the model to a dataset of *C. elegans* gap junction connectivity, demonstrating its potential in different neuronal systems. This extension serves to illustrate the model’s adaptability and potential applicability to a broader range of neuronal connectivity studies.

(3) The proposed method relies on the availability of neuronal connectomic data for model training,which may be limited or absent in certain brain connectivity settings.

We acknowledge the limitations posed by the model’s dependency on comprehensive connectomic data, which may not be readily available across all research contexts. To address this, we have discussed in the revised manuscript several alternative strategies to adapt our model to the available data. This includes exploring the potential of applying the model to available data such as projectome, and integrating other data modalities such as electrophysiological measurements. These initiatives aim to enhance the model’s applicability and ensure its utility in a broader spectrum of brain connectivity studies, especially in scenarios where detailed connectomic data are not available.

**Reviewer 2 (Recommendations For The Authors):**
Q1. In this work, the author has mainly been studying the retina neuronal type connectivity, it will be interesting to see whether the model works for other brain regions or other neuronal type connectivity as well.

We value your interest in the model’s applicability to other brain regions and neuronal types. To address this, we have extended our analysis in the revised manuscript to include a study on gap junction connectivity between *C. elegans* neurons. This extension demonstrates the model’s versatility and its potential applicability across various nervous systems and connectivity types.

Q2. Whether the authors can use the same transformation matrices trained from the retina data to predict neuronal connectivity in other brain regions? Or an easier case, the connectivity between RGC types to the neuronal types in SC, dLGN, or other post-RGC-synaptic brain regions. As the neuronal connection mechanisms are conserved and widely shared between different neuronal types, one would expect the same transformation matrices may work in predicting other neuronal type connectivity as well (at least to some extent).

The idea to use the same transformation matrices for predicting connectivity in other brain regions is intriguing. While direct application of these matrices to different regions remains challenging, we discussed the potential scalability of our model to other brain areas. By applying the model to combined datasets from various regions, we could uncover conserved neuronal connection mechanisms. This approach is theoretically feasible and is supported by the demonstrated scalability of the bilinear model and its deep learning variants in industrial applications.

Q3. Section 5.2 Connectivity metric generation: in this work, the author uses the stratification profiles of the neurons to estimate the connectivity metric, how reliable this method is? There will be a scenario where though two neuronal types project to a similar inner plexiform layer, they may not have any connection. Have the authors considered combining other experimental data (like electrophysiology data or neuron tracing data)?

We discussed the reliability of using stratification profiles for estimating connectivity metrics, acknowledging potential limitations. In the revised manuscript, we added discussion on how the integration of additional experimental data, such as electrophysiological and neuron tracing data, could enhance the accuracy of the connectivity metrics.

Q4. Section 6 Model training and validation: does the author have a potential hypothesis as to why 2 dimensions are the best latent feature spaces dimensionality? One would imagine with more dimensionality, the model will give better results. Could it be that the connectivity data that is used to train the model is only considering the two-dimensional space of the neuronal stratification?

The selection of two dimensions for the latent feature space was informed by 5-fold cross-validation, aimed at optimizing model generalization to unseen data. Here while increasing dimensionality improves performance on the training set, it does not necessarily enhance generalization to the validation set. Thus, the choice of two dimensions ensures good performance without overfitting to the training data.

Q5. Could the author provide the source code for the analysis? Or could the author make it a python/R package so that non-computational biologists can easily apply the method to their own data?

We have included a "Data and Code Availability" section in the revised manuscript. This section provides a link to the source code with pointers to datasets used in our study, facilitating the application of our methods by researchers from various backgrounds.

Q6. I know it may be difficult for the author to do, but is it possible to design and perform some experiments to validate the model prediction results, either connectivity partners of transcriptomicallydefined RGC types or the function of the key genetic molecules (which hasn’t been discovered before)? The author may consider collaborating with some experimental labs. The author may even consider predicting the connectivity between RGC with some of its post-synaptic neurons in the brain regions, like SC or dLGN, as recently there are a lot of single-cell sequencing data as well as connectivity data.

We appreciate your suggestion regarding experimental validation. As a future direction, we have discussed potential experimental approaches to validate the model’s predictions in the "Experimental Validation of Candidate Genes" section. Specifically, we propose an experimental design involving the manipulation of delta-protocadherins using AAV-mediated CRISPR/Cas9 and subsequent examination of connectivity phenotypes. We are also open to collaborating with experimental labs to further explore the model’s predictions, particularly in predicting connectivity between RGCs and their post-synaptic neurons in other brain regions.